# Large spread in the representation of compound long-duration dry and hot spells over Europe in CMIP5

Colin Manning[1], Martin Widmann[2], Douglas Maraun[3], Anne F. Van Loon[4], Emanuele Bevacqua[5]

[1]School of Civil Engineering and Geosciences, Newcastle University, Newcastle upon Tyne, United Kingdom
[2]University of Birmingham, Edgbaston, Birmingham, B152TT, United Kingdom
[3]Wegener Center for Climate and Global Change, University of Graz, Graz, Austria
[4]Institute for Environmental Studies (IVM), Vrije Universiteit Amsterdam, Amsterdam, the Netherlands
[5]Department of Computational Hydrosystems, Helmholtz Centre for Environmental Research—UFZ, Leipzig, Germany

*Correspondence to*: Colin Manning (colin.manning@newcastle.ac.uk)

**Abstract.** Long-duration, sub-seasonal, dry spells in combination with high temperature extremes during summer have led to extreme impacts on society and ecosystems in the past. Such events are expected to become more frequent due to increasing temperatures as a result of anthropogenic climate change. However, there is little information on how long-duration dry and hot spells are represented in global climate models (GCMs). In this study, we evaluate 33 CMIP5 GCMs in their representation of long-duration dry spells and temperatures during dry spells. We define a dry spell as a consecutive number of days with daily precipitation less than 1mm. CMIP5 models tend to underestimate the persistence of dry spells in Northern Europe while a large variability exists between model estimates in Central and Southern Europe where models have contrasting biases. Throughout Europe, we also find a large spread between models in their representation of temperature extremes during dry spells. In Central and Southern Europe this spread in temperature extremes between models is related to the representation of dry spells, where models that produce longer dry spells also produce higher temperatures, and vice versa. Our results indicate that this variability in model estimates is due to model differences and not internal variability. At latitudes between 50-60ºN, the differences in the representation of persistent dry spells are strongly related to the representation of persistent anticyclonic systems, such as atmospheric blocking and sub-tropical ridges. Furthermore, models simulating a higher frequency of anticyclonic systems than ERA5, also simulate temperatures in dry spells that are between 1.4 K, and 2.8 K warmer than models with a lower frequency in these areas. Overall, there is a large spread between CMIP5 models in their representation of long-duration dry and hot events that is due to errors in the representation of large-scale anti-cyclonic systems in certain parts of Europe. This information is important to consider when interpreting the plausibility of future projections from climate models and highlights the potential value that improvements in the representation of anticyclonic systems may have for the simulation of impactful hazards.

# 1 Introduction

The persistence of anticyclonic systems such as atmospheric blocks and sub-tropical ridges can lead to the co-occurrence of long-duration dry spells with extremely high temperatures in Europe. Such events have resulted in severe impacts across the continent. For example, the events of 2012 and 2018 led to extremely low crop yields (Kovačević et al., 2013; Beillouin et al., 2020) which resulted in agricultural insured losses of US$2 billion in Serbia in 2012 (Zurovec et al., 2015), while in 2018, financial support was required by farmers from governments in Sweden (€116 million), Germany (€340 million) and Poland (€116 million) (D'Agostino, 2018). Such events, characterised by the combination of multiple drivers causing extreme impacts, are known as compound events (Zscheischler et al., 2018; Zscheischler et al., 2020; Bevacqua et al., 2021). Anthropogenic climate change is expected to influence compound events (Seneviratne et al., 2012; Zscheischler et al., 2018; Seneviratne et al., 2021, Mukherjee and Mishra, 2021; Ridder et al., 2022), and so future planning for such changes requires reliable climate models that can represent these hazards, their combination and their underlying drivers. Despite this importance, studies evaluating climate model representation of compound events are still rare (Bevacqua et al., 2019; Zscheischler and Seneviratne, 2017; Zscheischler et al., 2021; Villalobos-Herrera et al., 2021; Ridder et al., 2021). In this article, we assess how well general circulation models (GCMs) from CMIP5 represent long-duration dry and hot events, as well as the influence of blocking on these events, over Europe during June, July and August (JJA).

Sub-tropical ridges are poleward extensions of the subtropical high-pressure belt into the middle and high latitudes (Sousa et al., 2021), while blocking anticyclones are large-scale, quasi-stationary anticyclones that block or divert the zonal westerly flow in the midlatitudes (Kautz et al., 2022). Both can occur in the life cycle of an anticyclonic system and previous studies have highlighted their local influence on the development of dry and hot conditions. The presence of anticyclonic conditions suppresses rainfall (Santos et al., 2009; Sousa et al., 2017) and increases the likelihood of dry spells persisting (Röthlisberger and Martius, 2019). These conditions are also conducive to the development of temperature extremes in summer (Meehl and Tebaldi, 2004; Cassou et al., 2005; Quesada et al., 2012; Stefanon et al., 2012; Tomczyk and Bednorz, 2016; Sousa et al., 2018) through increased incoming solar radiation (Pfahl and Wernli, 2012) and adiabatic warming due to subsidence (Zschenderlein et al., 2019; Nabizadeh et al., 2021) which cause temperatures to rise throughout an event (Miralles et al., 2014; Folwell et al., 2016). The presence of dry and hot conditions can subsequently deplete soil moisture levels (Teuling et al., 2013; Manning et al., 2018) and, in turn, amplify temperature extremes through land-atmosphere feedbacks (Seneviratne et al., 2010). Altogether, the above leads to an increased probability of extremely high temperatures during a dry spell (Manning et al., 2019).

CMIP5 models have been separately evaluated in terms of their representation of blocking, duration of dry spells and extreme temperatures. They generally struggle with the representation of blocking and underestimate its frequency (Scaife et al., 2010; Anstey et al., 2013; Masato et al., 2013; Dunn-Sigouin and Son, 2013; Davini and D'Andrea, 2016; Davini and D'Andrea,

2020; Schiemann et al., 2020). Similarly, CMIP5 models tend to underestimate both the annual number of dry days with precipitation below 1 mm (Polade et al., 2014) as well as the maximum duration of dry spells over much of Europe (Sillmann et al., 2013; Lehtonen et al., 2014). High temperatures are also underestimated over Europe, except in eastern areas (Sillmann et al., 2013; Cattiaux et al., 2013; Di Luca et al., 2020). These biases are likely inherited by model errors in the representation of blocking. For instance, Maraun et al. (2021), who found an underestimation of dry spell lengths over Austria in an ensemble of high-resolution models, show that it is partly explained by an underestimation in the persistence of the relevant synoptic weather types. Similarly, in an analysis of a smaller climate model ensemble, Plavcová and Kyselý (2016) showed that models simulating more persistent anticyclonic conditions tend to have longer heat waves.

Despite model errors in the representation of blocking (or anticyclonic systems), the linkage between heat waves and blocking is well simulated by climate models and blocking remains an important driver of temperature extremes in future climate simulations (Brunner et al., 2018; Schaller et al., 2018; Chan et al., 2022; Jeong et al., 2022). The linkage of such systems with dry spells, however, has not been assessed in CMIP5. It is therefore important that we understand how well this link is represented and whether or not errors in blocking have any repercussions for the representation of dry spells. Such information may help understand the plausibility of future projections of long-duration dry and hot events.

This study evaluates the ability of 33 GCMs from the CMIP5 ensemble to represent long-duration dry and hot events compared to observations. We assess the variability between models in their representation of such events and aim to understand possible reasons for the spread between models. Such spread between models can arise from internal climate variability as well as differences in model formulation that can influence their ability to represent certain processes. Within the analysis, we distinguish between these two sources in order to highlight reasons behind the spread. In particular, we are interested in understanding the extent to which biases in the representation of large-scale anticyclones can explain biases in the representation of long-duration, dry and hot events. For example, do models that simulate a higher blocking frequency also simulate longer and hotter dry spells?

## 2 Data

We employ daily maximum temperature and daily accumulated precipitation from the EOBS dataset (Haylock et al., 2008) version 16.0 between 1976 and 2005. We also obtain geopotential height data at 500hPa (Z500) from the ERA5 reanalysis dataset (Hersbach et al., 2020), also between 1976 and 2005. Daily maximum temperatures and daily precipitation accumulations were obtained for 33 climate models within the coupled model intercomparison project 5 (CMIP5) for simulation years from 1976 to 2005. However, Z500 could only be sourced for 25 models on a daily timescale. All data was regridded to a 2.5° by 2.5° lat-lon grid using the remapcon operator from the Climate Data Operators code (Schulzweida,

2009). Each model has a varying number of initial condition ensemble members (between 1 and 10) used to investigate internal variability. See Supplementary Table 1 for model details.

## 3 Methods

### 3.1 Dry Spells and Extreme Temperatures

The duration of a dry spell ($D_{DS}$) is defined as the number of consecutive days with precipitation below 1 mm. Only dry spells longer than 5 days are considered. The dry day threshold is consistent with previous studies and allows for comparison between observations and climate models which systematically overestimate the number of drizzle days ((Orlowsky and Seneviratne, 2012); (Donat et al., 2013); (Lehtonen et al., 2014); (Pfleiderer et al., 2019)). To compare temperatures during dry spells between models and with observations, we calculate the mean of the maximum daily-maximum-temperature during a dry spell ($Tx_{DS}$).

To quantify the relationship between temperatures and dry spells, we assess whether the odds (i.e. the probability of an event divided by the probability of a non-event) of a hot day is enhanced during a dry spell. Specifically, we calculate an odds ratio ($OR_{HD,n}$) as:

$$OR_{HD,n} = \frac{P_{HD,n}/(1-P_{HD,n})}{0.05/(1-0.05)}, \tag{1}$$

where $P_{HD,n}$ is the probability of exceeding a hot day threshold during a dry spell lasting longer than $n$ days (we consider dry spell durations ranging within $n = 5\text{-}20$ days). The hot day threshold is defined as the 95th percentile of the distribution of all daily temperatures during JJA for a given model and location, and 0.05 is the climatological probability. Values above 1 indicate that the odds of a hot day are increased during a dry spell that exceeds a specified duration. We also assess if the $OR_{HD,n}$ value at a given location can be achieved by random chance. To do so, we shuffle annual blocks of the precipitation series 1,000 times to provide 1,000 synthetic series of precipitation. By shuffling annual blocks, and not the daily values, we conserve the serial correlation of daily precipitation and the seasonality of dry spells. For each synthetic series, we calculate $OR_{HD,n}$ and estimate the upper bound of the 95% confidence interval, which is the 95th percentile of the 1,000 synthetic $OR_{HD,n}$ values. $OR_{HD,n}$ is deemed significant if it is greater than this upper bound.

### 3.2 Objective Detection of Anticyclonic Systems

A large number of indices have been developed to detect blocking, owing to the diverse range of synoptic patterns that the term 'blocking' refers to ((Barriopedro et al., 2010); (Barnes et al., 2012); (Woollings et al., 2018)). Different algorithms detect

different physical characteristics of blocks and can produce varying blocking climatologies (Pinheiro et al., 2019). It is therefore important to consider the nature of a given algorithm when interpreting results. We apply an algorithm developed by Sousa et al. (2021), which builds on a commonly used algorithm developed by Tibaldi and Molteni (1990). Ideally, it is favourable to compare results from multiple algorithms, and although this is beyond the scope of this current work, we do compare results produced by the Sousa et al. (2021) and Tibaldi and Molteni (1990) algorithms to demonstrate any sensitivities to algorithm choice.Sousa et al. (2021)(Tibaldi and Molteni, 1990)

The Sousa et al. (2021) algorithm uses daily mean geopotential heights at 500hPa (Z500) and is designed to delineate between structurally different anticyclonic features that have in the past been considered under the same blocking term, namely sub-tropical ridges, omega blocks and rex blocks. A sub-tropical ridge is defined as a poleward extension of the subtropical high, termed the subtropical belt, and generally exhibits an open pressure contour. In contrast an omega block exhibits a closed contour but remains attached to the subtropical belt, while a Rex block, which also has a closed contour, is generally cut-off from the subtropical belt and separated by a cyclonic system in between. In a conceptual model outlined by Sousa et al. (2021), the life cycle of an anti-cyclonic system generally comprises a sub-tropical ridge at the beginning and develops into an omega and/or rex block in the mature phase of the system. The algorithm from Sousa et al. (2021) builds on that first proposed by Tibaldi and Molteni, (1990), which detects blocking features, by adding the detection of subtropical ridges as well as differentiating between the above features. It therefore has the advantage in that it captures a larger proportion of the life cycle of anti-cyclonic systems than the original blocking algorithm would capture alone. It is also relatively simple to apply and uses a low number of parameters. While a detailed explanation of the algorithm and its rational is given in Sousa et al. (2021), we provide an overview of the steps required below which included local detection of ridges and blocking as well as spatial criteria.

### 3.2.1. Local Detection of Ridges and Blocking

A ridge is identified as a poleward extension of the subtropical belt into middle and high latitudes. Its detection firstly requires the identification of the sub-tropical belt which is defined each day separately as areas where the local Z500 value is higher than $\overline{[Z500]}$: the hemisphere-wide mean Z500, averaged over the previous 15 days preceding each day. Next, ridges within the subtropical belt are identified as areas with latitudes greater than $LAT_{MIN}$, which is the minimum latitude at which a subtropical ridge can occur on a given day. To calculate $LAT_{MIN}$ each day, the poleward edge of the subtropical belt is found at all longitudes as the maximum latitude at which a Z500 is greater than $\overline{[Z500]}$ at each longitudinal row. $LAT_{MIN}$ is then the average of these maximum latitudes.

Local and instantaneous blocking is identified using a 2D version of the Tibaldi and Molteni (1990) method. The algorithm identifies blocked grid cells as those with meridional flow reversals using geopotential height (Z500) gradients (GHG). Two gradients are calculated to the north (GHGN) and south (GHGS) of a given grid cell at longitude $\lambda$, latitude $\phi$, on day $d$:

$$GHGN(\lambda,\phi,d) = \frac{Z500(\lambda,\phi+\Delta\phi,d) - Z500(\lambda,\phi,d)}{\Delta\phi} \tag{2}$$

$$GHGS(\lambda,\phi,d) = \frac{Z500(\lambda,\phi,d) - Z500(\lambda,\phi-\Delta\phi,d)}{\Delta\phi} \tag{3}$$

Where $\Delta\phi = 15^o$ is a typical latitudinal extension of blocking. A block is identified at a given grid cell if GHGN < 0 m/degree latitude and GHGS > 0 m/degree latitude. Typically, a threshold of GHGN < -10 is used, but due to recommendations from Tyrlis et al. (2021), this has been relaxed. We have tested the sensitivity of results to this choice and find it has little influence on the overall results (not shown).

### 3.2.2. Application of Spatial Filter and Area Criteria

Further criteria are applied to remove unwanted features and ensure the detected ridge or block is a large-scale, spatially contiguous high-pressure system. After applying the local criteria outlined above, a spatial filter is applied to remove jet structures with strong winds that can surround ridges and blocks, ensuring we only keep grid cells embedded within the high-pressure system. The filter removes grid cells with GHG > 20 m/degree. GHG is a local measure of geostrophic wind magnitude where the wind magnitudes are inferred from zonal and meridional Z500 gradients calculated using centred differences of $\Delta\phi/2$ width in longitude and latitude, respectively. Next, all grid cells north of $LAT_{MIN}$ that have been identified as a ridge or block are grouped under the same classification. For each day, only grid cells that are grouped within spatially contiguous structures with at least a 500,000 km$^2$ areal extent are kept.

The application of the criterion $LAT_{MIN}$ means that grid cells below this latitude on a given day are excluded and this results in little or no detection of systems at latitudes below 40ºN during summer. Hence, most locations in Southern Europe including the Iberian Peninsula, Italy and the Balkans have little to no occurrences of anticyclonic systems as defined here. To ensure only physically meaningful results are included, we exclude grid cells below 40ºN from the analysis related to anticyclonic systems. Further criteria may be applied to delineate between the different types of structures (e.g. ridges, omega block, rex block). We do not apply such criteria and prefer to classify all ridge and block systems under the same term, Anticyclonic Systems (AS), as both can occur within the same event and also exbibit the same local influence on rainfall and temperatures. However, for completeness, we also assess results produced when only the local blocking criteria from the Tibaldi and Molteni

(1990) method (Eq. 2 and Eq. 3) are considered (presented in supplementary information). This will include all grid cells including those below 40°N and provide an indication of the sensitivity of results to the choice of criteria used to detect anticyclonic systems.

## 3.3 Quantifying Influence of Anticyclonic Systems on Dry Spell Persistence

We quantify the relationship between the persistence of anticyclonic systems (AS) and of dry spells (DS) following the
205 approach of Röthlisberger and Martius (2019), who studied the influence of blocking on dry spells. The climatological persistence of $k$-type spells (i.e., AS spell or DS) at grid point $g$ can be quantified by calculating the climatological (daily) survival probability ($Ps_{g,k}$) as:

$$Ps_{g,k} = P\big(Spell_{g,k}(t+1) = 1 \,\big|\, Spell_{g,k}(t) = 1\big), \tag{4}$$

where $t$ refers to a daily timestep, $k$ indicates either AS or DS, and $Spell_{g,k}$ is a binary variable where 1 indicates a dry day for dry spells and an anticyclonic day for when an anticyclonic system is present. To assess the effect of anticyclonic systems on dry spell persistence, the survival probability of dry spells when an anticyclonic system is present is calculated as:

$$Psa_{g,DS} = P\big(Spell_{g,DS}(t+1) = 1 \,\big|\, Spell_{g,AS}(t) = 1 \cap D_{AS}(t) \geq 5\big), \tag{5}$$

where $D_{AS}(t)$ indicates the total duration of the anticyclonic system that overlaps with this day. $Psa_{g,DS}$ therefore represents the survival probability of a dry spell when it co-occurs with an anticyclonic system whose total duration is at least 5 days. In a next step, the odds of a dry spell surviving when an anticyclonic system is present, $Psa_{g,DS}/(1 - Psa_{g,DS})$, are compared
with the climatological survival odds of dry spells, $Ps_{g,DS}/(1\text{-}Psa_{g,DS})$ by calculating an odds ratio (OR):

$$OR_{DS} = \frac{Psa_{g,DS}/(1 - Psa_{g,DS})}{Ps_{g,DS}/(1 - Psa_{g,DS})}, \tag{6}$$

The value of $OR_{DS}$ indicates how the odds of dry spell survival change when an AS spell is present at the same time. For
example, a value greater than one indicates that the AS spell enhances the dry spell survival probability. This approach demonstrates the relationship between anticyclonic conditions and the day-to-day persistence of dry spells.

### 3.4 Estimation of Duration Return Levels

We estimate return levels (RLs) for the duration of dry spells that have an estimated return period (RP) of 5 years. We choose to look at RLs with a RP of 5 years so that we focus on dry spells that may be impactful but also frequent enough to draw robust conclusions.

RLs are estimated using a parametric approach in which we fit an exponential distribution to the duration of all dry spells and anticyclones that exceed 5 days. The use of the exponential distribution is common for modelling the probability of dry spells (Serinaldi et al., 2009; Manning et al., 2019). The RL ($d$) for a RP ($T$) of $n$ years is estimated as:

$$d = F^{-1}\left( 1 - \frac{\mu}{T} \right), \tag{7}$$

where $F^{-1}$ is the inverse of the fitted cumulative distribution function (CDF) and $\mu$ is the exceedance rate, calculated as $\mu = \frac{N_E}{N_Y}$, where $N_E$ is the number of dry spells exceeding a duration of 5 days and $N_Y$ is the number of years.

### 3.5 Calculation of Metrics and Regional Analysis

For a given metric, prior to computing the multi-model median, we calculate the ensemble mean for each model individually. This ensures that each model has equal weighting in the calculation of multi-model median metrics. The median is used instead of the mean for the CMIP5 ensemble as this better represents of the centre of the multi-model ensemble, as it is not influenced by an outlier model. We also present regional results in order to summarise results across the CMIP5 ensemble. For each model, metrics are averaged across three IPCC European regions (Northern Europe, Central Europe, and Southern Europe) as defined by Seneviratne et al. (2012). The separation between the regions is shown by black dashed lines in Figure 1c.

### 4 Results

### 4.1 Representation of Long-Duration Dry Spells in CMIP5 Models

The return level (RL) for the duration of a dry spell with an expected return period of 5 years across Europe is presented for EOBS (Fig. 1a) and the multi-model median of the 33 CMIP5 models (Fig. 1b). The spatial distribution of RLs based on EOBS (Fig. 1a) is in line with documented differences in synoptic variability across Europe. That is, persistent anticyclonic conditions in the south favour longer dry spells than over northern Europe, where shorter durations are in line with a higher synoptic variability between cyclonic and anticyclonic conditions (Ulbrich et al., 2012). The persistent anticyclonic conditions in the south arise from the subtropical high sitting over Southern Europe in summer (Sousa et al. 2021) and can lead to dry spells lasting the entire summer.

The spatial variability of RLs in southern and northern Europe is well captured by the CMIP5 multi-model median (Fig. 1b). However, the mean relative difference between EOBS and CMIP5 (Fig. 1c) indicates that CMIP5-based 5-year RLs can be shorter than those from EOBS (blue grid cells) by 30-50% across a large area of Europe including Scandinavia, Western Central Europe and the Iberian Peninsula. It is particularly the case in Scandinavia, where more than 90% of models show
shorter 5-year RLs than EOBS, as indicated by the stippling. In contrast, CMIP5 based 5-year RLs in the south-eastern part of the domain are higher than those from EOBS. Boxplots in Fig. 1d show the variability between models of the 5-year RLs averaged across each of the IPCC regions. The boxplots reflect the results in Fig. 1c, particularly in Northern Europe where CMIP5 models tend to produce shorter 5-year RLs. The results in Central and Southern Europe vary more across the models as they tend to simulate both lower and higher RLs. The spread across the CMIP5 ensemble is also quite high with differences
between models and EOBS ranging from 20% shorter to 60% longer. The interquartile range is higher in Central and Southern Europe than in Northern Europe while the overall variability is highest in Southern Europe.

The differences between EOBS- and CMIP5-based RLs can arise from internal variability within climate realisations and from differences between model formulations. To understand the sources of these differences, we compare the regional means of
the 5-year RLs for all ensemble members of each model. Figure 2 shows that the typical spread between members within each model ensemble is smaller than the spread across all CMIP5 models (top row). This indicates that the spread across the CMIP5 ensemble (Fig. 1d) is very likely due to differences in model formulations and not internal variability. This result and the spread between models (Fig. 1d) points to errors of models in the CMIP5 ensemble in capturing the climatology of long-duration dry spells. It can therefore be expected that, for many models, future projections of dry spells and associated variables
such as temperature and soil moisture are also not fully realistic.

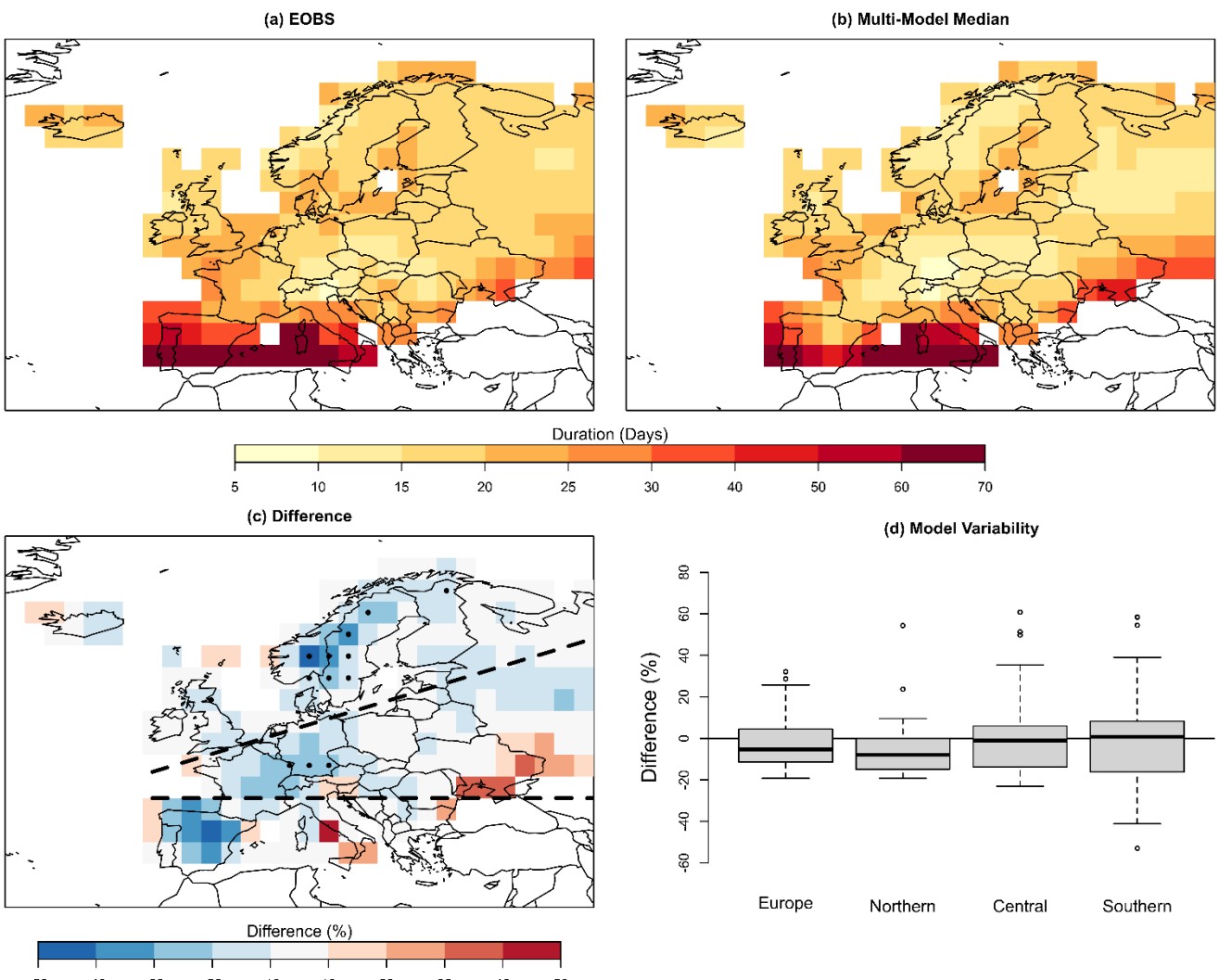

**Figure 1:** Duration Return Levels (RLs) of dry spells for a 5-year return period for (a) EOBS, and (b) the median of the CMIP5 multi-model ensemble. (c) Percentage difference between CMIP5 multi-model median and EOBS (stippling indicates where 90% of CMIP5 models are below or above EOBS). (d) Model spread in the relative difference averaged across all grid cells in Europe, Northern Europe, Central Europe and Southern Europe (dashed lines in (c) indicate the three European IPCC regions).

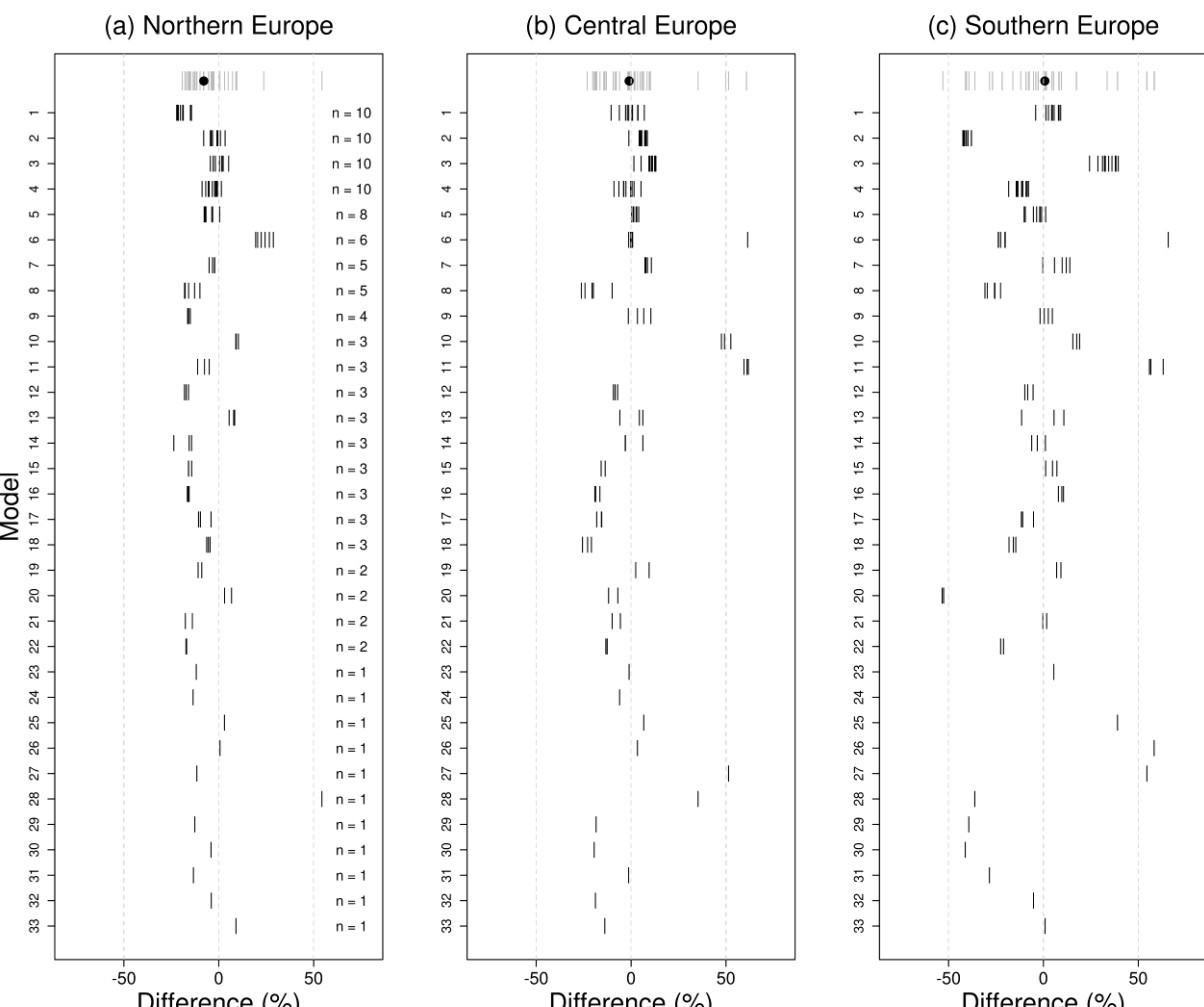

**Figure 2:** Relative difference in Duration RLs (model - EOBS) calculated for all members of each model ensemble in three regions: (a) Northern Europe; (b) Central Europe; and (c) Southern Europe. First row provides the ensemble mean of each model (grey lines) and the multi-model ensemble median (black dot), while each subsequent row provides the relative difference for each ensemble member of models 1-33 and the number of members (n) in each model ensemble.

**4.2 Representation of Temperature During Dry Spells**

The mean of the maximum temperatures during dry spells exceeding 5 days ($Tx_{DS}$) for EOBS and the CMIP5 multi-model median is presented in Fig. 3. The spatial pattern of temperature seen in EOBS is generally reproduced by CMIP5 though, as also shown in Cattiaux et al. (2013), underestimations of $Tx_{DS}$ are seen across most of Europe (Fig. 3c,d). The majority of models show an underestimation in $Tx_{DS}$ in both Northern and Southern Europe though the models in Central Europe have contrasting biases in this region (Fig. 3d), while Central Europe also has the largest model spread in $Tx_{DS}$. The largest differences between the multi-model median and EOBS are generally found in coastal areas. This may be a result of the regridding process as sea temperatures may be included for the models. Hence, biases in these areas may not be as meaningful as those further inland. To test if the coastal differences are likely to influence the spread seen in the ensemble, we calculate the inter-model standard deviation at each grid cell (Fig A1). The standard deviations at coastal grid cells are not higher than the nearby grid cells in land. In fact, the highest standard deviations are found further inland away from the coast. As model variability is not higher around the coast, it is therefore unlikely that the spread between models (Fig 3d) is due to this coastal effect.

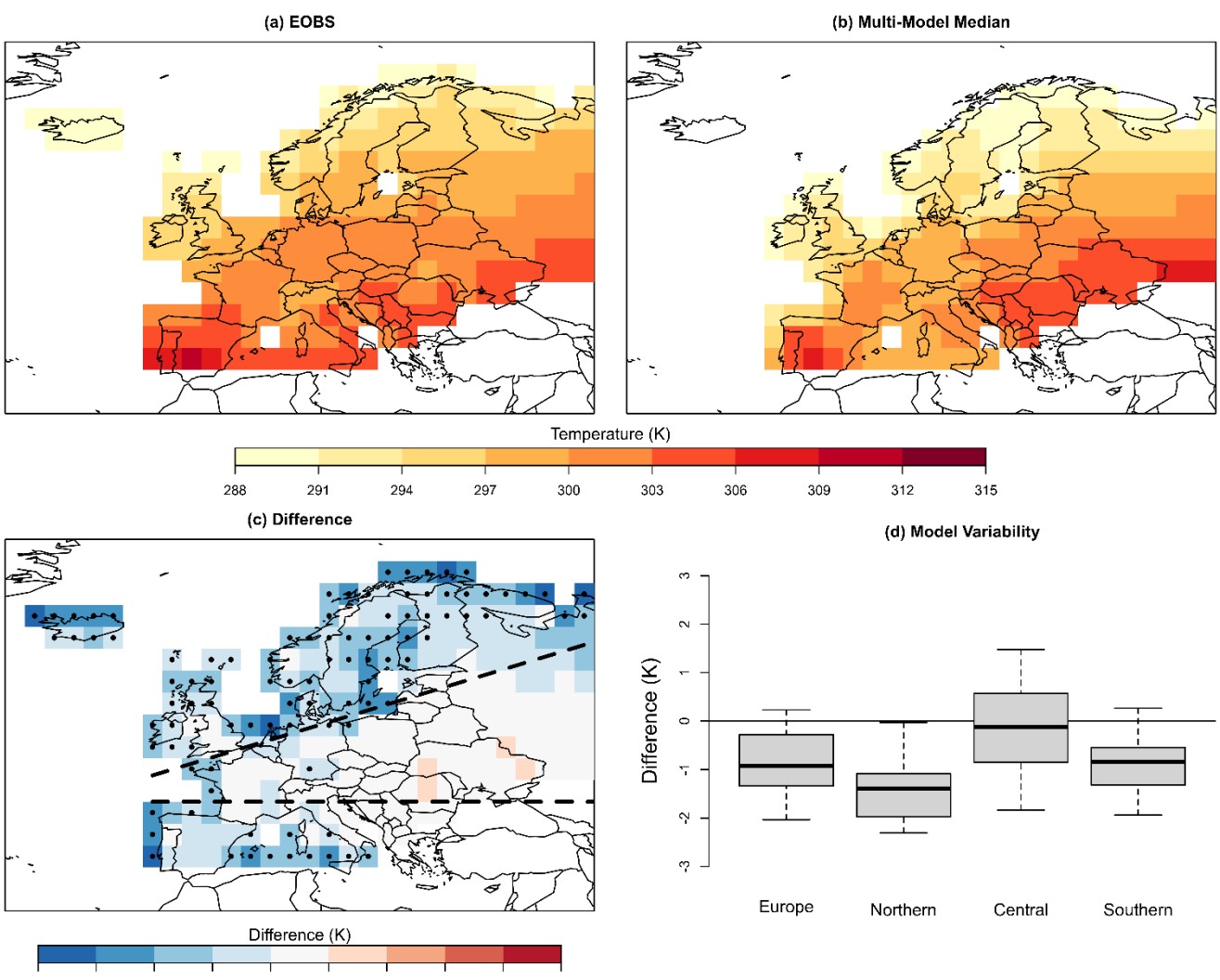

**Figure 3:** Mean maximum temperatures during dry spells longer than 5 days ($Tx_{DS}$) in (a) EOBS and (b) the CMIP5 multi-model median. (c) Multi-model median difference between CMIP5 models and EOBS (stippling indicates where 90% of CMIP5 models are below or above EOBS). (d) The variability in the percentage difference across all models averaged across all grid cells in Europe, Northern Europe, Central Europe and Southern Europe. The separation between the three European regions is shown by the dashed lines in (c).

We also assess whether the differences between models are more likely due to internal variability or from systematic differences in model formulations. In Fig. 4, we compare the regional means of $Tx_{DS}$ for all ensemble members of each model. Similarly to dry spell durations, we also see that the spread in the differences between members within each model ensemble

is quite low and much less than the spread across the CMIP5 ensemble (top row). This indicates that the spread across the CMIP5 ensemble is largely due to inherent model differences and not internal variability.

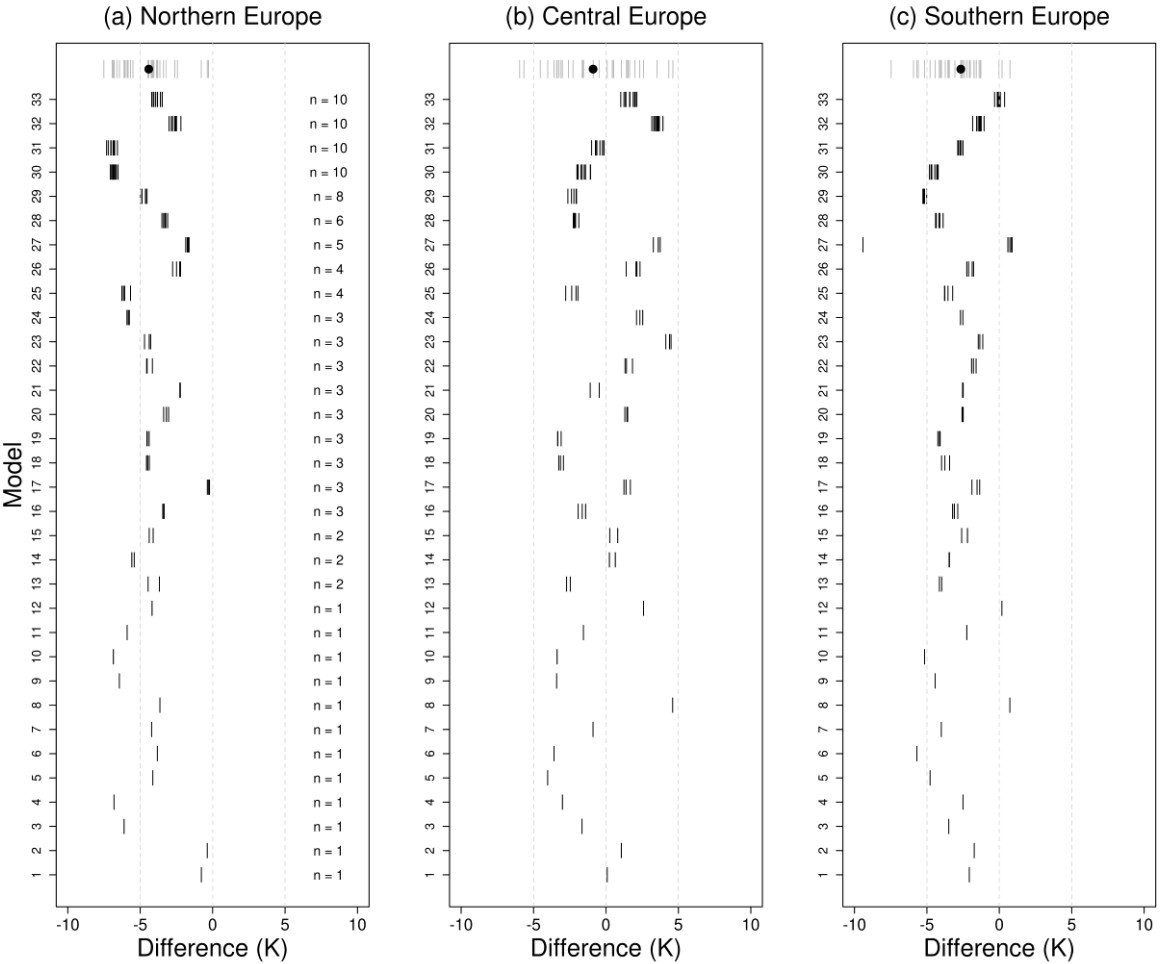

**Figure 4:** Difference in $Tx_{DS}$ (model - EOBS) calculated for all members of each model ensemble in three regions: (a) Northern Europe; (b) Central Europe; and (c) Southern Europe. First row provides the ensemble mean of each model (grey lines) and the multi-model ensemble median (black dot), while each subsequent row provides the differences for each ensemble member of models 1-33 and the number of members (n) in each model ensemble. Models are sorted by number of members

335 in descending order.

## 4.3 Relationship between Temperature Extremes and Dry Spells

In the EOBS dataset, there is an increased probability of temperature exceeding its 95[th] percentile during dry spells that last longer than 5 days (Fig. 5a). Stippling, which is present across a large area of Europe, indicates that we are 95% confident that the results cannot be achieved via random chance at those locations. The highest ratios in EOBS are seen in northwestern Europe, where ratios > 2 indicate that the odds of temperature exceeding the 95[th] percentile is more than doubled during a dry spell that is longer than 5 days, compared to the odds when considering all days. Across the rest of Northern, Central, and Southeastern Europe, ratios generally vary between between 1.25 and 2. In parts of Southern Europe, the ratios vary around 1 and there is a lack of stippling. This is a consequence of the high number of dry days there during summer as dry spells can last entire summers (Fig. 1). Hence, the potential for coupling between dry spells and temperatures in Southern Europe is less, and the closer the total number of dry days is to the total number of summer days, the closer the odds ratio will be to 1. The spatial variability in the odds ratio thus reflects differences in the degree of coupling between dry spells and temperature which is likely due to differences in drivers of dry spells and temperature extremes across Europe. In more Northern parts with higher synoptic variability, dry spells and temperature extremes are both driven by, and linked to, the synoptic variability of anticyclonic systems (Röthlisberger and Martius, 2019). In Southern Europe, where the subtropical high persists for large parts of summer, dry conditions are the norm throughout such that dry spells and temperature extremes vary independently there. Hence, the odds ratio results should be interpreted with caution, requiring careful consideration of the number of dry days at a given location.

The spatial variability of the odds ratio is well captured by the CMIP5 multi-model median (Fig. 5b) though over- and under-estimations are evident in parts of France and Northern Europe. Figure 5c-e shows the spread between models and the sensitivity of the estimated ratio to the duration of dry spell. The ratio is calculated for dry spells exceeding 1 to 20 days and then averaged across the three regions. For EOBS in Central and Northern Europe, the ratio increases with increasing duration up to 10 days and levels off at around 2, although there is likely to be some spatial variation in the ratio as shown in Fig. 5a. In Southern Europe, the ratio remains close to 1 and increases slightly after 10 days. The CMIP5 multi-model median ratio shows a similar pattern to EOBS in that it increases with increasing dry spell duration and is generally quite comparable in magnitude. However, the CMIP5 ensemble shows considerable spread in the estimated odds ratio, particularly in Central and Southern Europe. The spread is largest for the longest durations which is likely a sampling issue as the number of dry spells decreases with the increasing duration threshold.

The relevance of differences in the odds ratio between models is difficult to interpret. An under- or over-estimation can indicate that temperature extremes coincide with long dry spells less or more often than in observations respectively. Both of which may have different implications for impacts. However, this interpretation is complicated by the fact that the odds ratio is influenced by the number of dry days at a given location. Hence, models with a higher number of dry days are more likely to

have a smaller ratio, and vice versa. Overall, the results give an indication that the models generally capture the observed relationship between dry spells and temperature, as they compare well spatially (Fig. 5a,b) and capture the increased probability of extreme temperatures during longer dry spells (Fig. 5c-e).

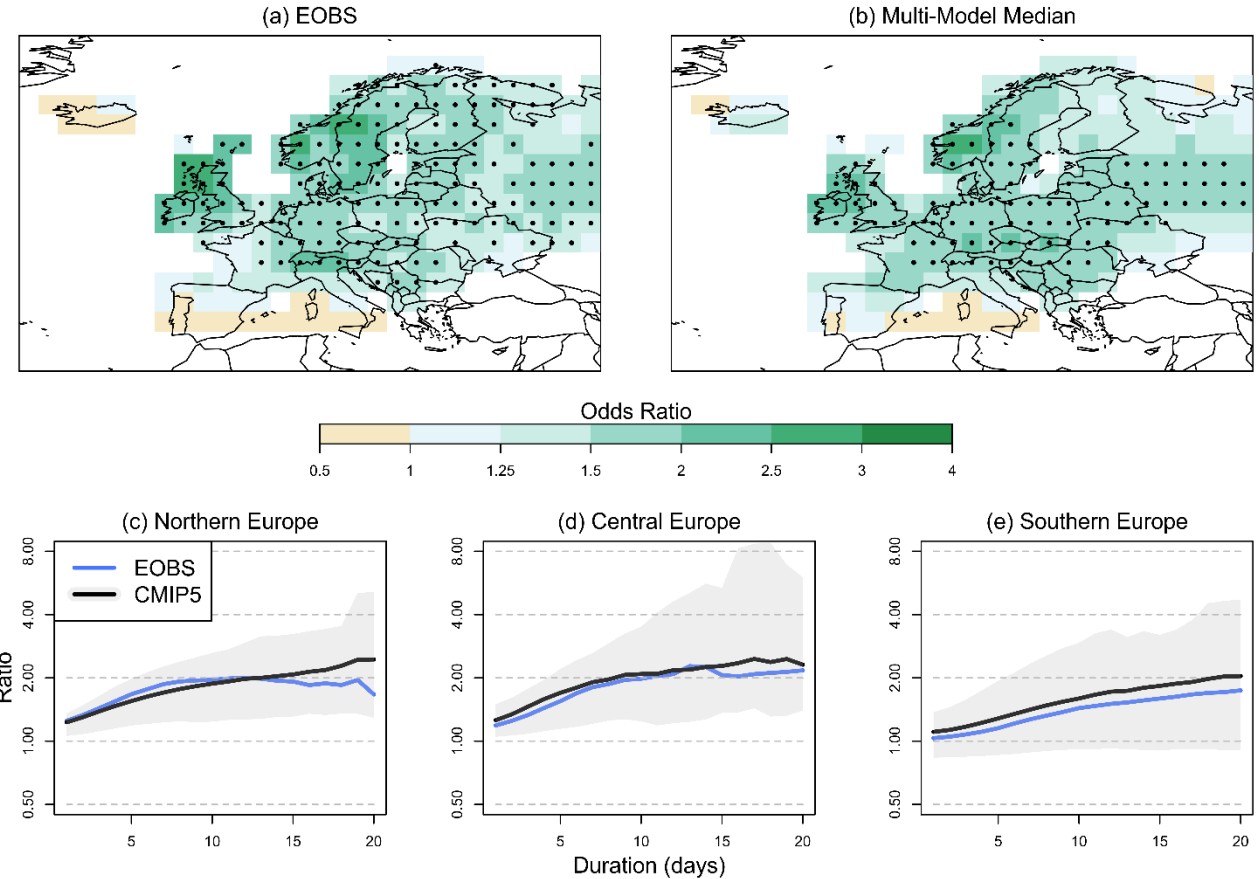

**Figure 5:** Comparison of the relationship between dry spells and temperature quantified as the odds ratio ($OR_{HD,n}$) (see section 3.1) in (a) EOBS and (b) the CMIP5 multi-model median. Stippling indicates that there is a less than 5% probability that the odds ratio can be achieved by random chance. Only dry spells longer than 5 days are included. Sensitivity of the odds ratio to the duration of dry spell averaged across (c) Northern Europe, (d) Central Europe and (e) Southern Europe for EOBS (blue line) and the CMIP5 multi-model median (solid black line). The grey area represents the model spread in the ratio.

**4.4 Relationship between Temperature and Dry Spell Duration Biases**

In this section we assess the relationship between dry spell duration and temperature biases and compare models in terms of their joint ranking in their representation of these two components. To do so, we calculate the inter-model Spearman correlation coefficient between $RL_{DS}$, the 5-year RLs for the duration of dry spells, and $Tx_{DS}$, the average of the maximum temperature (Fig. 6). A positive inter-model correlation is found between $RL_{DS}$ and $Tx_{DS}$ over a large area of Central and Southern Europe (Fig. 6a) while there is generally little correlation between them in Northern Europe. Positive correlations indicate that models that simulate longer dry spells tend to produce higher extreme temperatures. This is particularly the case over Central European countries such as France and Germany where correlations vary between 0.5 and 0.9.

The points in the scatter plots shown in Fig. 6b-d provide the areal mean $RL_{DS}$ and $Tx_{DS}$ values over the three European regions (the grey dot in each panel represents the EOBS values to illustrate how models differ from EOBS). The figure gives an overview of the relationship between the the differences in the representation of long-duration dry and hot events. A large spread exists between the models, particularly in Central and Southern Europe where the positive relationship is seen between $RL_{DS}$ and $Tx_{DS}$. The climatology of events in CMIP5 models ranges from shorter-cooler events to longer-hotter events, particularly in Central Europe where the variability in $RL_{DS}$ is much higher than that seen in the rest of Europe. From an impact perspective, models with longer-hotter dry spells indicate a higher compound event risk, or at least the expected impacts from a simulated climate with shorter-cooler events may be much different to those in a simulated climate with longer-hotter events. In the next section, we investigate the extent to which the representation of large-scale anticyclonic systems can explain this spread.

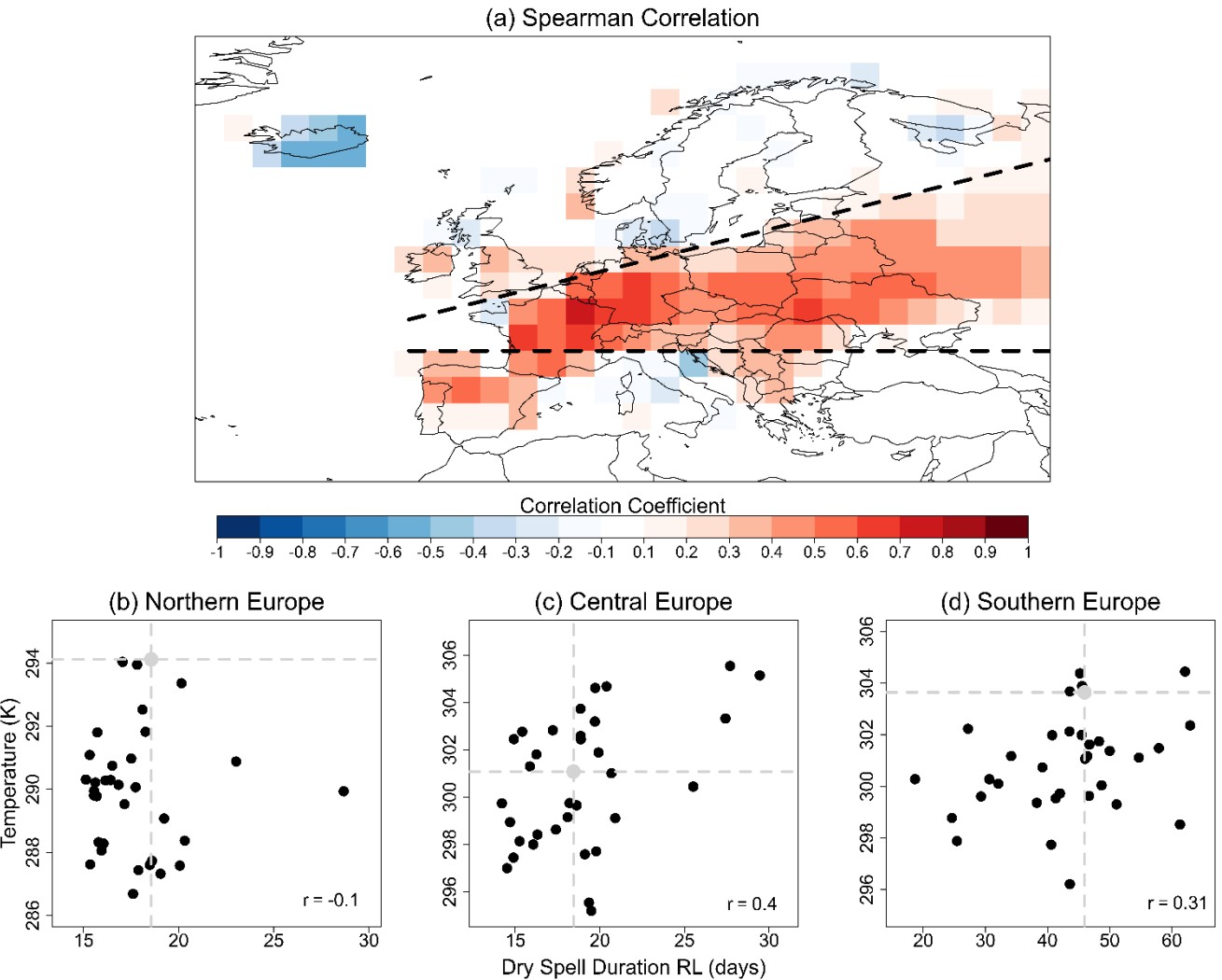

**Figure 6:** Relationship between $RL_{DS}$, the 5-year RLs for the duration of dry spells, and $Tx_{DS}$, the average of the maximum temperature during dry spells longer than 5 days. (a) Inter-model Spearman correlation coefficient. Panels (b), (c) and (d) show the inter-model relationship between $RL_{DS}$ vs. $Tx_{DS}$ averaged across (b) Northern Europe, (c) Central Europe, and (d) Southern Europe for each CMIP5 model. The Spearman correlation coefficient calculated from the 33 models is provided in the bottom left corner of each panel. The three IPCC regions (Seneviratne et al., 2012) are indicated by the black dashed lines in panel (a).

## 4.5 Anticyclonic Systems: Frequency and Influence on Dry Spells

The frequency of anticyclonic systems (AS) across Europe, according to the Sousa et al. (2021) algorithm, is presented for ERA5 (Fig. 7a) and for the multi-model median of the 25 CMIP5 models (Fig. 7b) for which daily Z500 data was available. The analysis is only shown for grid cells north of 40ºN as the algorithm filters out AS occurring at lower latitudes within the subtropical high (see methods section 3.2) which can persist over Southern Europe throughout summer resulting in very long dry spells (Fig. 1). The spatial distribution of AS frequency in ERA5 (Fig. 7a) is in line with that already shown in Sousa et al. (2021), though differences are present, likely due to the different time period considered here. A high frequency is found just north of 40ºN, which is largely due to the frequent presence sub-tropical ridges there (Sousa et al., 2021). Frequencies decrease with increasing latitude in the north where highest frequencies are found over Scandinavia. The CMIP5 multi-model median captures this spatial variability (not shown) though differences exist in the absolute frequencies (Fig. 7b). In line with previous studies (e.g. Antsey et al., 2013; Masato et al., 2013; Dunn-Sigouin and Son, 2013; Davini et al., 2021), the multi-model median underestimates AS frequency derived from ERA5 across Northern Europe, as well as in western Europe. In contrast, the multi-model median shows similar or higher frequencies across Eastern Europe. The spread between models is discussed later alongside the spread in dry spell durations.

The presence of anticyclonic conditions increases the likelihood of a dry spell persisting. The odds ratios ($OR_{DS}$) presented in Fig. 7c,d show whether a dry spell is more likely to persist for another day when it co-occurs with an anticyclonic system lasting at least 5 days. The survival probability of dry spells in EOBS is increased at most locations across the domain (where $OR_{DS} > 1$), and everywhere in central and northern Europe, when co-occurring with an anticyclonic system, though there are spatial variations. Lowest values are found over parts of Central Europe close to the Mediterranean near alpine areas, while largest values (> 3) are found across Northern Europe. This spatial variability indicates that dry spell persistence in Northern Europe is more reliant on synoptic conditions than in Central Europe where other factors such as moisture availability, convective systems, and topography may play a role. The spatial variation in the CMIP5 multi-model median (Fig. 7b) is similar to that in EOBS though the magnitude of the relationship is underestimated over large parts of Europe, particularly in parts of Scandinavia and Central Europe.

Given the link between AS and dry spells seen in observations and in the models, we now assess the variability of AS frequency in CMIP5 models and whether this can explain the variability seen in the duration of dry spells with a 5-year RL ($RL_{DS}$), as well as that in the average of maximum temperatures seen during dry spells longer than 5 days ($Tx_{DS}$). The inter-model Spearman's rank correlation coefficient is calculated between AS frequency and $RL_{DS}$ (Fig. 8a), as well as between AS frequency and $Tx_{DS}$ (Fig. 8b). High positive correlations (>0.6) are seen across much of Northern Europe between 50-60ºN. These areas generally coincide with the areas that have a high odds ratio (Fig. 7b). Similarly, positive though weaker correlations of 0.4 are found between AS frequency and $Tx_{DS}$ at these latitudes. These relationships, and the variability of AS

frequency between models, are further illustrated using scatter plots (Fig. 8c-h). For all models, we compute the areal mean of each metric within three regions highlighted by black boxes in Figure 8a: UK & Ireland (UK&I), Central Europe, and Eastern Europe. The scatter plots reflect the correlations in Figure 8a and also show that models underestimating AS frequency compared to ERA5 (grey dot) also underestimate the 5-year RL of dry spell durations from EOBS (Fig. 8c-e), and vice versa. Furthermore, the scatter plots indicate the presence of a non-linear relationship between AS frequency and $Tx_{DS}$ over each

region, particularly the UK and Ireland and central Europe. In these regions, models with blocking frequencies higher than ERA5 have a higher $Tx_{DS}$. For instance, the average $Tx_{DS}$ for models with a higher AS frequency than ERA5 (points to the right of the vertical line in Figure 8f-h) is 1.4 K, 1.8 K and 2.8 K warmer than models with a lower AS frequency over UK&I, central Europe and eastern Europe respectively.

The results demonstrate the strong constraint that the representation of anticyclonic conditions have for the persistence of long-duration dry spells, and to a lesser extent for the magnitude of temperatures within them between latitudes 50-60ºN. Hence, in these areas, models with systematic biases in AS frequency will also misrepresent the persistence of dry spells and contribute to biases in temperature. Outside 50-60ºN, little or no correlation is found. It is unclear why low correlations are found in other parts of Europe, particularly Scandinavia. It is possible that non-local effects of anticyclonic systems may play role, in that

high AS frequencies in one location may lead to wetter conditions in areas surrounding the system, while other sources of biases may play a larger role such land-atmosphere interactions. It is also possible that a different algorithm may yield different results. For example, we have repeated the same analysis using blocking criteria only from the Tibaldi and Molteni (1990) method (see Eq. 2 and 3), presented in Figure A2. Results are generally similar to those above though we do see positive correlations between blocking frequency with both dry spell duration and temperature over larger areas of France and Southern

Europe, although the correlations are weaker than those seen in more northern areas. However, we also see reduced correlations with temperature over Eastern parts of Europe and over the UK and Ireland. The results highlight that care may be needed when choosing an algorithm for the detection of anticyclonic systems. The optimum choice may depend on the region of interest. Lastly, we note that the results shown in Fig. 8 are insensitive to reasonable changes in a number of parameters of the AS algorithm that were tested (GHGN, GHGS, $LAT_{MIN}$ and AS duration).

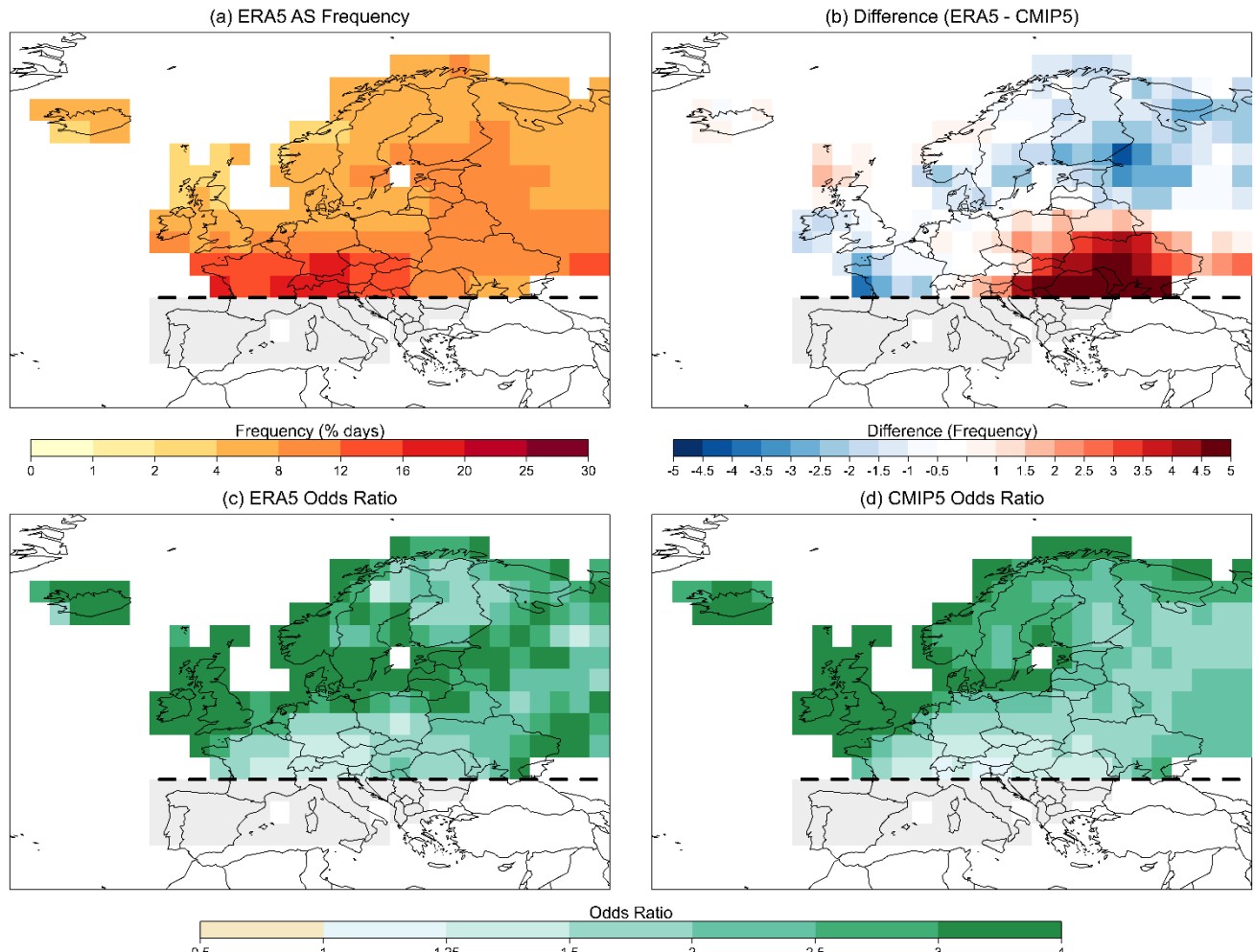

**Figure 7:** The frequency of anticyclonic systems (AS) according to the algorithm from Sousa et al. (2021) in (a) ERA5 and (b) the difference between ERA5 and the CMIP5 multi-model median. Odds Ratios ($OR_{DS}$) for dry spells when co-occurring with an anticyclonic system lasting at least 5 days in (c) EOBS and (d) CMIP5. Grey areas indicate the masked grid cells below 40ºN marked by the dashed black line.

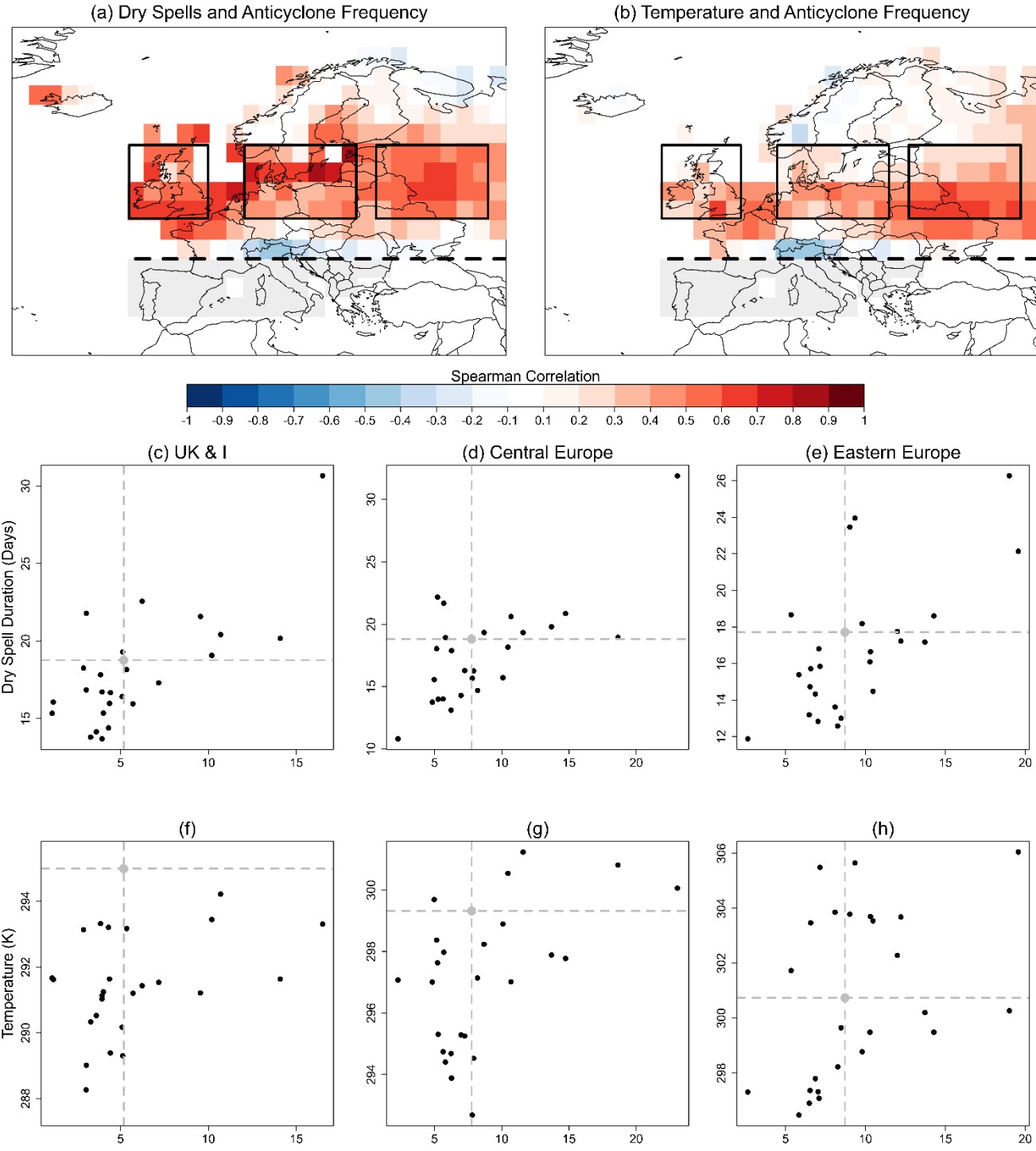

**Figure 8:** Relationship between frequency of anticyclonic systems (AS), that last for at least 5 days, with 5-year RLs of dry spell durations ($RL_{DS}$) and with the average maximum temperature from dry spells longer than 5 days ($Tx_{DS}$). (a) Inter-model Spearman's rank correlation coefficients between (a) AS frequency and $RL_{DS}$, and (b) AS frequency and $Tx_{DS}$. Scatter plots show inter-model relationships between AS frequency and $RL_{DS}$ averaged over (c) UK & Ireland, (d) Central Europe, and (e) Eastern Europe; as well as AS frequency and $Tx_{DS}$ averaged over (f) UK & Ireland, (g) Central Europe, and (h) Eastern Europe. Each point represents a model while the grey dot in each panel represents the metrics obtained from EOBS ($RL_{DS}, Tx_{DS}$) and ERA5 (AS frequency). The three regions used to demonstrate these relationships are indicated by the black boxes in panel (a). Grey areas indicate the masked grid cells below 40ºN marked by the dashed black line.

## 5 Discussion

This paper evaluates the representation of long-duration dry and hot events over Europe in the CMIP5 ensemble. The aim of the paper was to demonstrate the variability between models in their representation of such events and to understand possible reasons for this spread between models. In particular, we are interested in understanding the extent to which biases in the representation of large-scale anticyclones can explain biases in these events.

The duration of dry spells is calculated as the consecutive number of days with precipitation less than 1 mm. Our findings are consistent with previous analyses of CMIP5 (e.g. Polade et al. 2014; Sillman et al., 2013; Lehtonen et al., 2014). In Northern Europe, CMIP5 models tend to underestimate the 5-year return level for the duration of a dry spell while there are contrasting differences between models in Central and Southern Europe where some models underestimate and others overestimate the 5-year return level. These model differences are found to be due to inherent differences in model formulations and not internal variability as the typical spread across ensemble members of a single model is substantially smaller than the overall spread across the CMIP5 ensemble. Similarly, we assessed models in their representation of temperatures during dry spells. Specifically, we calculated the mean of the maximum temperatures from all dry spells longer than 5 days and find that temperature extremes are underestimated in Northern and Southern Europe while contrasting differences are seen in Central Europe. There is also a large spread between models throughout Europe and our results indicate that this spread arises from differences in model formulations rather than internal variability. We acknowledge that many models have few members within their ensemble (Table A1) and so the CMIP5 ensemble is not the best tool to separate contributions to the spread from internal variability and model differences. Single model initial-condition large ensembles (SMILES) may offer a better alternative to explore this question (Maher et al., 2021; Bevacqua et al., 2022), though it is still important to assess this within multi-model ensembles such as CMIP5.

The relationship between the above biases in dry spell durations and temperatures was assessed by calculating the inter-model

Pearson correlation coefficient between the 5-year return level in dry spell durations and the mean of the maximum temperatures from dry spells longer than 5 days. This revealed a strong positive relationship in Central Europe, and a positive but weaker correlation in Southern Europe, meaning that models that simulate longer dry spells also simulate higher temperatures, and vice versa. The reasoning for this relationship is likely related to land-atmosphere interactions which have an important influence on both temperature and precipitation in this region (Seneviratne et al., 2010). Climate models have

difficulty in accurately simulating soil moisture as well as the partitioning between latent and sensible heat fluxes at the land surface which can contribute to precipitation and temperature biases (Dong et al., 2022). However, the direction of causality of biases is not straightforward and biases arising from atmospheric drivers may amplify those driven by soil moisture. For instance, long dry spells could deplete soil moisture which may in turn increase temperatures ((Mueller and Seneviratne, 2014); (Berg et al., 2015); (Lin et al., 2017)). Similarly, warmer models may deplete soil moisture more leading to reduced moisture

recycling, less precipitation, and longer dry spells (Vogel et al., 2018).

The representation of persistent anticyclonic conditions may also modulate both the representation of duration and temperature of dry spells. We have assessed the influence that biases in anticyclonic systems (AS) have on the representation of the duration of dry spells and temperatures within them. To do so, we applied an algorithm from Sousa et al., (2021) to identify AS. This

algorithm detects a range of anticyclonic features including atmospheric blocking and sub-tropical ridges. With this we have assessed the representation of AS frequency as well as their influence on dry spell persistence in observations and models. In line with previous papers that have assessed blocking frequency (Antsey et al., 2013; Masato et al., 2013; Dunn-Sigouin and Son, 2013; Davini and D'Andrea, 2016; Davini and d'Andrea, 2020; Schiemann et al., 2020), AS frequency is underestimated in much of Northern Europe by the majority of models, though there are a few that simulate higher frequencies. Despite this,

models generally represent the link between AS and dry spells that is seen in observations. Specifically, we demonstrate in observations and models, that the odds of a dry spell lasting another day is almost 4 times higher in much of Northern Europe when it co-occurs with an AS, as has previously been shown in Röthlisberger and Martius (2019) for observations. This result is similar to Brunner et al. (2018) who demonstrate the link between blocking and extreme temperatures is realistically represented in a climate model despite its underestimation in blocking.

Following this, we computed the inter-model Spearman's rank correlation coefficient between AS frequency and the 5-year return level in dry spell durations and find high positive correlations at latitudes between 50-60°N. Hence, a model that underestimates AS frequency will also underestimate the persistence of dry spells at these latitudes, and vice versa. Positive correlations are also found in these areas between AS frequency and the average maximum temperatures during dry spells,

$Tx_{DS}$. The latter correlations are much weaker though there is evidence to suggest a non-linear relationship exists between a model's simulation of AS frequency and $Tx_{DS}$. For example, the average $Tx_{DS}$ for models with a higher AS frequency than

ERA5 is between 1.4 K, and 2.8 K warmer than models with a lower AS frequency in these areas. South of 50ºN, we see little correlation with AS frequency. This is likely due to criteria used in the Sousa et al. (2021). For instance, we also assess a simpler algorithm (Tibaldi and Molteni, 1990) that identifies blocking anticyclones only (Fig A2), rather than the combination of sub-tropical ridges and blocking features (Fig. 8). From this algorithm, we see strong positive correlations over France, and weakly positive correlations in Southern Europe likely a result of the low frequency of blocking there (not shown) compared to the long dry spells that can persist for the entire season due to the subtropical high. Both algorithms produce similar results though with some regional differences. Hence, care should be taken when choosing an appropriate algorithm and the choice may depend on the region of study. Finally, we note that we have only assessed the summer season in this analysis, and so different results may be found for other seasons. This may particularly be the case in winter and spring in Southern Europe when synoptic variability is higher than in summer due the sub-tropical high sitting further to the south (Sousa et al., 2021). For example, blocking played a large role during Spring 2004 in the development of a major drought over the Iberian peninsula (García-Herrera et al., 2007).

## 6 Conclusion

The results reveal a large spread in the representation of long-duration dry and hot events within the CMIP5 ensemble. This is largely due to differences in the representation of persistent large-scale anticyclonic systems at latitudes between 50-60ºN. In central parts of Europe, it is possible that biases in dry spell durations lead to temperature biases, or vice versa, likely through land-atmosphere interactions. Given that biases in these events arise through errors in the large-scale circulation and in the representation of the land-surface, a performance-based constraint on model selection (e.g. McSweeney et al., 2015; Vogel et al., 2018; Brunner et al., 2020) or a process-based analysis of plausible future extremes is likely required (e.g. Fischer et al., 2021) when assessing the current and future risk posed by long-duration dry and hot events. Particularly as blocking is shown to remain important for heat waves in both present and future climates (Brunner et al., 2018; Schaller et al., 2018; Chan et al., 2022, Jeong et al., 2022). This multivariate perspective is also important for impact modelling studies (Zscheischler and Seneviratne, 2017), which require bias adjustment procedures to create more usable input data. These methods have their limitations (Doblas-Reyes, 2021) and are not designed to correct for large-scale errors (Maraun et al., 2017). Ideally, studies employing methods such as those that simply correct dry day frequencies (e.g. Hempel et al., 2013; Samaniego et al., 2018) should also consider a models' performance in the relevant atmospheric processes. Otherwise, unintended consequences may arise such as increasing biases in the modelled impact (Zscheischler et al., 2019) or breaking the relationship between drivers, such as the large-scale circulation, and the hazard of interest (Addor et al., 2016; Maraun et al., 2021).

In summary we have shown that climate model biases in the frequency of anticyclonic systems have repercussions for the representation of dry spells and temperatures within dry spells during summer. The relationships between biases imply that

improvements in the representation of anticyclonic systems can be expected to lead to improvements in the representation of dry spells and temperatures. Improvements in blocking have already been reported in the CMIP6 ensemble (Schiemann et al., 2020) and it would therefore be interesting to test if the expected improvements in dry spells and temperature can also be seen.

**Code and data availability.** All data assessed is publicly available and was accessed from the CEDA archive via the JASMIN supercomputer. Code can be made available upon reasonable request.

**Author Contributions.** Concept was developed by CM, EB, MW, and DM. CM undertook the analysis and preparation of the manuscript. All authors contributed to the interpretation of results and writing of the manuscript.

**Competing interests.** The authors declare they have no competing interests.

**Acknowledgements.** Colin Manning is supported by UKRI NERC funded projects FUTURE-STORMS (NE/R01079X/1) and STORMY-WEATHER (NE/V004166/1). Colin Manning gratefully acknowledges the support he receives as a visiting
scientist from the UK Met Office Hadley Centre. Emanuele Bevacqua has received funding from the European Union's Horizon 2020 research and innovation programme under grant agreement no. 101003469.

# Appendix A: Additional Tables and Figures

**Table A1: CMIP5 models used in the analysis. The model IDs correspond to those in Figures 2 and 6. Models are arranged in descending order of ensemble size (N). A 'Y' in the Z500 column indicates that daily output for 500hPa geopotential heights were available for the specified model.**

| ID | Institute | Model | N | Z500 | ID | Institute | Model | N | Z500 |
|----|-----------|-------|---|------|----|-----------|-------|---|------|
| 1 | CCCma | CanCM4 | 10 | | 18 | NCC | NorESM1-M | 3 | Y |
| 2 | CNRM-CERFACS | CNRM-CM5 | 10 | Y | 19 | CSIRO-BOM | ACCESS1-0 | 2 | Y |
| 3 | CSIRO-QCCCE | CSIRO-Mk3-6-0 | 10 | | 20 | LASG-CESS | FGOALS-g2 | 2 | Y |
| 4 | MOHC | HadCM3 | 10 | Y | 21 | MPI-M | MPI-ESM-P | 2 | Y |
| 5 | ICHEC | EC-EARTH | 8 | Y | 22 | BNU | BNU-ESM | 1 | Y |
| 6 | IPSL | IPSL-CM5A-LR | 6 | Y | 23 | CMCC | CMCC-CESM | 1 | Y |
| 7 | CCCma | CanESM2 | 5 | Y | 24 | CMCC | CMCC-CM | 1 | Y |
| 8 | MOHC | HadGEM2-ES | 4 | Y | 25 | CMCC | CMCC-CMS | 1 | Y |
| 9 | NOAA-GFDL | GFDL-CM3 | 4 | Y | 26 | INM | inmcm4 | 1 | |
| 10 | BCC | bcc-csm1-1 | 3 | Y | 27 | IPSL | IPSL-CM5B-LR | 1 | Y |
| 11 | BCC | bcc-csm1-1-m | 3 | Y | 28 | NASA-GISS | GISS-E2-H | 1 | |
| 12 | CSIRO-BOM | ACCESS1-3 | 3 | Y | 29 | NASA-GISS | GISS-E2-R | 1 | |
| 13 | IPSL | IPSL-CM5A-MR | 3 | Y | 30 | NOAA-GFDL | GFDL-ESM2G | 1 | Y |
| 14 | MOHC | HadGEM2-CC | 3 | Y | 31 | NOAA-GFDL | GFDL-ESM2M | 1 | |
| 15 | MPI-M | MPI-ESM-LR | 3 | Y | 32 | NSF-DOE-NCAR | CESM1-BGC | 1 | |
| 16 | MPI-M | MPI-ESM-MR | 3 | Y | 33 | NSF-DOE-NCAR | CESM1-CAM5 | 1 | |
| 17 | NCAR | CCSM4 | 3 | Y | | | | | |

**Ensemble Standard Deviation**

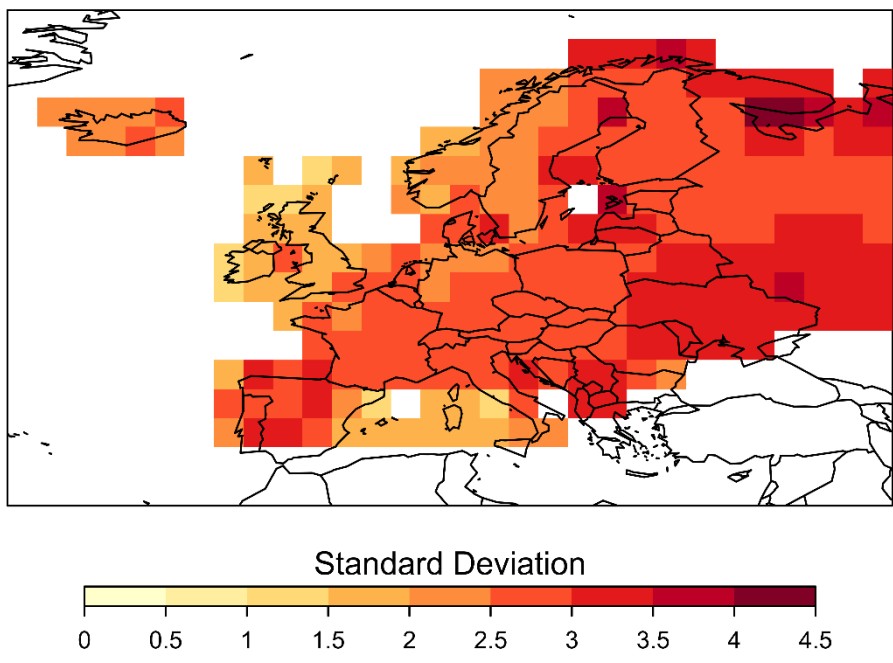

**Figure A1: Inter-model standard deviation in Tx$_{DS}$, the average of the maximum temperature during dry spells longer than 5 days.**

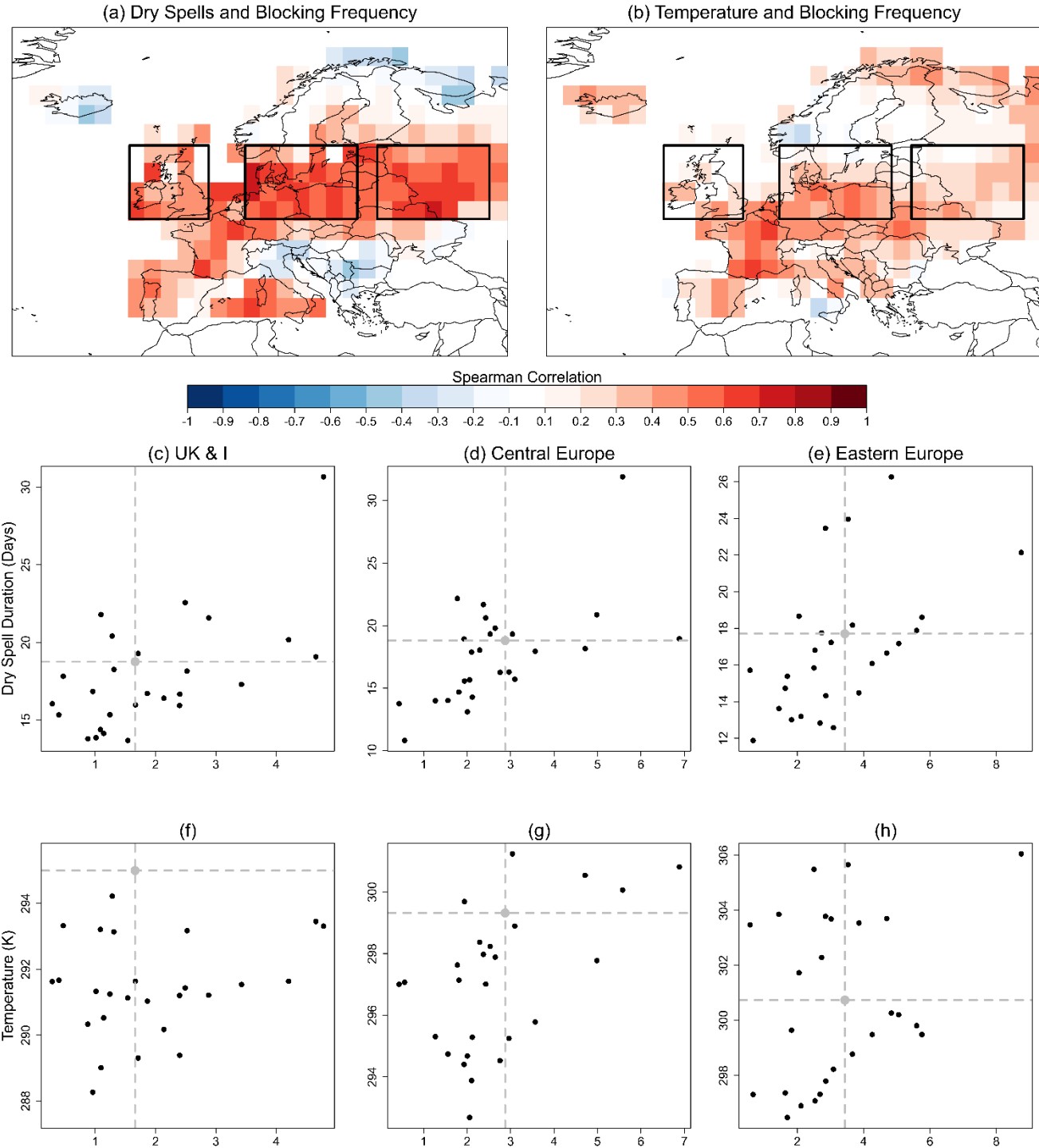

**Figure A2:** Relationship between frequency of blocking anticyclones, that last for at least 5 days, with 5-year RLs of dry spell durations ($RL_{DS}$) and with the average maximum temperature from dry spells longer than 5 days ($Tx_{DS}$). (a) Inter-model Pearson correlation coefficients between (a) Blocking frequency and $RL_{DS}$, and (b) Blocking frequency and $Tx_{DS}$. Scatter plots show inter-model relationships between AS frequency and $RL_{DS}$ averaged over (c) UK & Ireland, (d) Central Europe, and (e) Eastern Europe; as well as Blocking frequency and $Tx_{DS}$ averaged over (f) UK & Ireland, (g) Central Europe, and (h) Eastern Europe. Each point represents a model while the grey dot in each panel represents the metrics obtained from EOBS ($RL_{DS}$, $Tx_{DS}$) and ERA5 (AS frequency). The three regions used to demonstrate these relationships are indicated by the black boxes in panel (a).

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
