# Peer review of "Large spread in the representation of compound long-duration dry and hot spells over Europe in CMIP5"

_Weather and Climate Dynamics, 2022_

## Author Response (AR1)

General Comments

The authors analyse the links between the occurrence of anticyclones, dry spells and heat waves in a large ensemble of CMIP5 GCM simulations for the historical period. Additionally, ERA5 and E-OBS data is considered in the evaluation. The authors conclude that the discrepancies between the GCMs identified in terms of the duration of dry spells and extreme temperatures are related with the GCM biases regarding temperature and precipitation themselves. While the topic of the manuscript is surely interesting and I acknowledge that there was a huge amount of data processing involved, there are also severe shortcomings, the main one being the way that "anticyclones" are considered, which is methodological not sound (see major point). Given that this was the only aspect broadly related with atmospheric dynamics (the core of WCD), I'd argue that the manuscript in its present form is also out of scope of the journal. For both reasons, and given that this shortcoming is pretty fundamental to the whole analysis, I'd like to recommend its rejection in the present form (as the whole calculations and analysis would need to be redone).

This say, I'd strongly encourage the authors to take up this task and resubmit the manuscript for further evaluation. If resubmitted to WCD, the aspect of atmospheric dynamics should be strongly strengthened so it fits the scope of the journal (e.g. also links to blocking, synoptic weather types, anticyclone dynamics). If the authors do not which to change the focus of the manuscript so strongly, I'd suggest the resubmission to a different journal – notably NHESS, which focus primarily on impacts. I would be willing to review the paper again upon resubmission.

Main Comment

The main shortcoming in the present study is the way "anticyclones" are dealt with. For me, an anticyclone is a high-pressure centre with clockwise rotating winds and large-scale divergence at the surface. At upper levels, it is typically associated with a cut off low / ridge / blocking system, where upper-level convergence occurs, thus leading to large-scale subsidence in the area of the surface high-pressure centre. As described e.g. in a recent review paper published in WCD (Kautz et al. 2022; https://doi.org/10.5194/wcd-3-305-2022), a persistent anticyclone / blocking over Central Europe in the summer typically leads to heat waves and dry spells collocated with the centre of the system (see their Figure 2b). However, on both flanks of the system you often observe heavy precipitation associated e.g. with moist air intrusions on the western flank of the system (same figure).

Given this, I am really puzzled that the authors "define an anticyclone" as local exceedance of MSLP above 1012 hPa over five days. This value is below the average mean MSLP for a considerable part of Europe (particularly in the summer months!), and even below the global average MSLP value! Given the often quite stationary weather conditions in the summer, five days is also no real constraint. So we are not even looking at above average pressure conditions. And of course, this has nothing to do with atmospheric dynamics and cannot provide any insight either on the exact location of the anticyclone and where different types of extremes may be expected. If a simple metric is needed, I'd use chose something based on MSLP anomalies to the monthly mean (or summer) MSLP fields, thus as an indication of the anomalous circulation associated with the high-pressure anomaly.

This explains why several of the following evaluations/results do not really meet the expectations (e.g. compared to what you'd expect for blocking, e.g. around lines 265) or are not really understandable (at least to me), e.g. Figs 3 and low collocation of dry spells and anticyclones for Southern Europe (as acknowledged by the authors also in the text), and several other figures, particularly Figure 8. While I do like the idea of the paper (which is my I accepted the review), I do thing the methodology is flawed and the presented results are thus unfortunately not sound.

We thank the reviewer for their time in reviewing this manuscript, it is much appreciated. We accept that the way we have treated anticyclonic conditions is fundamentally flawed and so we have updated the analysis and applied an algorithm from Sousa et al. (2021) that detects anticyclonic features, namely sub-tropical ridges and atmospheric blocking using geopotential heights at 500 hPa. We combine these features under the same definition, anticyclonic systems, as both will have the same local effect on precipitation and temperature and both can also occur within the same life cycle of an anticyclone. The updated analysis assesses the frequency of anticyclonic systems in Europe according to the chosen algorithm, their local influence on the persistence of dry spells and the link between model biases in the frequency of anticyclonic systems in models and biases in both dry spell persistence and the magnitude of temperatures during dry spells in models. The text has also been improved to include a greater discussion of the literature surrounding anticyclonic systems, their influence on dry spells and temperature, as well as implications of the results. We feel the analysis and strength of the conclusions around anticyclonic systems is much improved and we hope that the paper will provide a meaningful contribution to the discussion around the importance of assessing the influence of biases in the large-scale circulation on relevant hazards when using climate models to aid future decision making.

Changes have been made to the text throughout and new sections have been added related to the additional analysis of anticyclonic systems.

- **Description of the methodology to detect anticyclonic systems is described in section 3.2 (L126-189).**
- **The new results related to anticyclonic blocking are provided in section 4.5 (Figure 7 and Figure 8).**

Minor comments

a) It is not clear to me why version 16.0 of E-OBS is being used, as we are currently on version 24.0 (see https://www.ecad.eu/download/ensembles/download.php). There have been quite a few important updates since.

**The older version was used as this had been downloaded and processed as part of a project that officially finished a few years ago. Having read through the updates, we do not note any major update that would yield this version unusable and do not expect to see large differences in the results from this version and the latest version. We are confident that the results and conclusions are insensitive to this choice.**

b) I think it is a very strong statement to say that the "combination of dry spells and extreme temperatures" has not yet been assessed in CMIP5 models. Please weaken the statement.

**This statement has been weakened (L66).**

c) The description of the results is often not understandable, e.g. the description of figure 3b (likes 228-236). Please enhance.

**Thank you for pointing this out, we have enhanced the description of the results throughout.**

d) Several of the references are missing page ranges, issue numbers, particularly for AGU journals, please enhance.

**Thank you for checking this and we apologies for this mistake. All references have been updated to include this information in format required by the journal.**

This manuscript investigates the representation of compound dry and hot spells in Europe in the CMIP5 data set. The model data is compared to EOBS. The results show that CMIP5 models struggle to capture the duration and intensity of these compound events. The manuscript is well written, the figures are clear and the results are relevant. I recommend to publish the manuscript after major revisions as detailed below.

**We thank the reviewer for taking the valuable time to review this manuscript, it is much appreciated.**

Major points:

The choice of a constant MSLP threshold needs to be further motivated and discussed. There are several issues with this choice. I) heat lows can from over the Iberian peninsula (https://rmets.onlinelibrary.wiley.com/doi/abs/10.1256/qj.01.189) during hot conditions breaking the link between MSLP and high temperatures, while the overall tropospheric circulation is still anticyclonic. Ii) in locations with high orography the correction of the surface pressure to MSLP might introduce biases. Iii) the climatologically lower pressure at higher latitudes leads to longer exceedances over the 1012hPa threshold compared to lower latitudes.

**We accept that the way we have treated anticyclonic conditions is flawed and so we have updated the analysis and applied an algorithm from Sousa et al. (2021) that detects anticyclonic features, namely sub-tropical ridges and atmospheric blocking using geopotential heights at 500 hPa. We combine these features under the same definition, anticyclonic systems, as both will have the same local effect on precipitation and temperature and both can also occur within the same life cycle of an anticyclone. The updated analysis assesses the frequency of anticyclonic systems in Europe according to the chosen algorithm, their local influence on the persistence of dry spells and the link between model biases in the frequency of anticyclonic systems in models and biases in both dry spell persistence and the magnitude of temperatures during dry spells in models.**

**Changes have been made to the text throughout and new sections have been added related to the additional analysis of anticyclonic systems.**

- **Description of the methodology to detect anticyclonic systems is described in section 3.2 (L126-189).**
- **The new results related to anticyclonic blocking are provided in section 4.5 (Figure 7 and Figure 8).**

A direct comparison of absolute temperatures between EOBS and CMIP5 (Figure 5) will be strongly affected by the representation of the orography and coast lines within CMIP5. A comparison relative to a percentile might be more meaningful.

**We agree that the representation of the coastline and orography may play a role in the temperature biases. However, we do not understand the rational behind why a percentile-based approach would remove these effects as they will be seen across the temperature distribution. However, we have added text (L293-295) to note the influence of coastal effects on the interpretation of the biases in those locations.**

Please control for multiple testing in all analyses using the FDR (see Wilks 2916, https://journals.ametsoc.org/view/journals/bams/97/12/bams-d-15-00267.1.xml)

**The cited paper refers to parametric tests which require statistical assumptions. Our approach uses non-parametric bootstrapping which does not require such assumptions. In the approach, we randomly shuffle seasons to break the seasonal dependence between the precipitation and temperature series and calculate the metric. This is repeated 1000 times and provides an indication of whether the result can be achieved by random chance. As the approach is non-parametric, FDR does not apply here, and we interpret the presence of stippling as there being a < 5% chance of the result occurring by random chance.**

How relevant is the representation of summer convection in the models for the duration of the dry spells?

**This is an interesting question though one which we cannot answer here. It would likely require a detailed analysis of the models with specific types on convection parameterisation schemes and/or a comparison with a high-resolution convection permitting climate simulations.**

Minor points

Abstract: long-duration $if$ sub-seasonal (Long duration is per se not very clear, it could also refer to spells that last for several years)

**Thank you for the suggestion, this has been added to the abstract.**

 38 Zscheischler 2020/2021 is missing in the list of references

**We apologise for this omission; we have updated the reference list.**

Add Ridder et al. 2022 to the list of references https://www.nature.com/articles/s41612-021-00224-4

**Thank you for highlighting this paper, we have now cited this in the introduction (L45).**

96 Is the mean taken across all spells? The definition is not yet fully clear.

**Yes, we calculate the maximum temperature within each dry spell lasting longer than 5 days. $Tx_{DS}$ is then the average maximum temperature from all those dry spells.**

182 IPPC $if$ IPCC

**Thank you for noticing this mistake, it has been corrected.**

410ff Include the results of Zscheischler and Seneviratne (https://www.science.org/doi/full/10.1126/sciadv.17002639) in the discussion.

**This has been included in the discussion (Line 537).**

Figure 1a I recommend to use a colormap with only one color, two colors suggest a change in sign.

**Thank you for this suggestion, and we agree that it might appear as a change in sign though as the paper does not assess changes in the hazards, we feel this is not a large issue. The colormap has been chosen to remain consistent with a previous paper (Manning et al., 2019), and we chose this colormap as it highlights the large difference between dry spell lengths in Northern and Southern Europe. As such, we prefer to keep the current colormap.**

Panels c,d,e in Figure 3 do not fit the description and look the same as panels c,d,e in Figure 7, there may have been a mix-up.

**Thank you for pointing this out and we apologies for this mix up. This mistake has been corrected.**

---

## Author Response (AR2)

**Note: All Page (P) and Line (L) numbers refer to the manuscript version with track changes.**

**Editor's comments**

Dear Colin Manning and co-authors,

Thank you for enhancing the dynamical component of the paper, with the integration of the new identification of anticyclonic features, and their relation to dry spells and extreme temperatures. With this analysis the work is in the scope of WCD and is a timely contribution to the discussion on model circulation biases and their relation to compound extremes.

There are yet important points to consider, as suggested by the referees and myself here below, especially concerning the conclusions drawn for latitudes south of 40N, and the strong statements on large discrepancies/errors which are not fully backed up by the results shown. I will be looking forward to considering your revised manuscript and response to each of the points raised.

Kind regards,
Shira

**Thank you for your positive comments and for your time in considering our manuscript for publication. It is much appreciated. We have made changes to the paper in line with the comments received from the reviewers. Specifically, we have masked latitudes below 40N in the blocking analysis (Figures 7 and 8). We have also included a supplementary analysis using an algorithm that identifies blocking anticyclones only (i.e. subtropical ridge criteria are removed), and we have clarified our statements around what is meant by 'large discrepancies' and given more discussion around this point. Exact details in these changes are laid out below.**

- AS persistence: I was somewhat confused by the mixed usage of daily AS (Fig. 7a,b), and AS lasting for 5 days (Figs. 7c,d and 8). Are there discrepancies in the representation of AS that last for 5 days (as is claimed in the abstract and in line 531)? Is it different from the relatively good representation of daily AS shown in Fig. 7a,b? Also in line 531 you mention Northern Europe, and given the results in those figures I assume you refer to central Europe? Likewise, the statement on central Europe should rather refer to southern/northern Europe?

- **The daily AS persistence should have been for AS lasting 5 days to be coherent with the remaining analysis, as this is the event definition from Sousa et al. (2021). We have updated Figure 7a,b so that it compares persistence of AS that last 5 days. The results are similar to those for daily AS persistence in that we see a relatively good comparison in the spatial distribution, although CMIP5 persistence is lower over parts of Scandanavia and higher in Eastern central Europe. We have also only shown points over land to be consistent with all other figures in the paper. Furthermore, as per request of one of the reviewers, we have updated Figure 7b to show the difference between ERA5 and CMIP5. We have also changed the colour bar in Figure 7c,d to a more appropriate colour bar and changed the colour so that it is evident to the reader that the metric in Fig. 7c,d is different to those in Fig. 7a,b.**
- **When we referred to Northern Europe, we should have said 'latitudes at 50-60N'. This has been updated (P31, L598). The statement on central Europe is correct as this is where we see the positive inter-model correlation between dry spells and temperature (Figure 6).**

- Identification of dry spell events: it is expected that each region in Europe experiences different numbers of dry events lasting over 5 days, with the southern parts potentially covering the entire season given their long duration. This information is currently missing though, but as noted in the text, it is important for interpreting the association with temperature extremes. Please enhance this information, and potentially use it to explain your choice of including/omitting data south of 40N.

- **We have touched on this within the text (P8, L260-262) though we have added further text to enhance its explanation (P8, L262-264).**
- **We have decided to omit grid cells below 40N when using the Sousa et al (2021) method. This reasoning for this is described on P7, L191-195. However, for completeness, we also included the analysis when using blocking criteria only (compared to the blocking and ridge criteria used in the Sousa et al. (2021) method), as described on P6 L199-202. These results are shown in Figure A2 in the appendix and discussed on P24 L486-492 and P30 L573-580. This additional analysis provides an indication of the sensitivity of the results to the choice of algorithm used to detect anticyclonic systems.**

- For which 26 models is daily Z500 provided? This information can be added to Table 1.

- **This should have been 25 models and has been updated (P3, L100). Table A1 has also been updated to show the models that provide daily Z500.**

- Units for duration are missing from Fig. 5.

- **Thank you for pointing this out. Units have now been added to Fig. 5.**

- The term "anticyclonic spell" is not clear. Please change to "anticyclonic system" or "dry spell" where relevant.

- **This term has now been replaced (P7, L215-223)**

- Fig. 7c,d: it will be useful to change the colour scale such that values above 1 are distinct (e.g., flipping the scale so that OR between 0.5 and 1 is yellow).

- **We have changed the colour scale to a brown-green colour scale, where brown is assigned to values below one. We prefer a different colour scale here in order to highlight that a different metric is being shown in Fig. 7c,d compared for Fig. 7a,b. For coherency, we also change the colour scale for odds ratios in Fig. 5.**

- Line 424: the first "seen" should be replaced by "in".

- **This has been changed (P23, L464).**

- Fig. 8c-f: units of frequency (%) do not match the numbers.

- **Thank you for pointing this out. This has been corrected.**

Reviewer #1

**General Comments**

The authors have made considerable changes in the manuscript, while lead to a strong improvement. The consideration of the Sousa et al method is a strong point, and the effort taken by the authors is acknowledged. I have one major comment and a few minor points. If these are properly address, I think this paper may become a strong contribution to WCD.

**We thank the reviewer for their valuable time and effort taken to review our manuscript. This is much appreciated. Your constructive comments have certainly improved the manuscript.**

**Major Comment**

I think that the lack of correlation between high temperatures, dry periods and blocking over Southern Europe during summer (Figs 6-8) is simply due to the fact that these are the average conditions for the region and this time of year. For example, southern Iberia must have about 95% of dry days in July and August, temperatures above 40°C are common (except for coastal areas), and anticyclones / ridges dominate the synoptic picture for weeks. Moreover, the fact that the frequency of blocking is not defined south of 40°N (lines 183-184; Fig. 7) makes things even more complex. Given the above, no wonder that no linear correlations can be found for this area and these variables.

My suggestion would be either i) to disregard this region in the analysis (and focus on Central Europe, for which the results are interesting and meaningful), ii) to use another approach rather than linear correlation for the analysis, or iii) to properly flag the reasons for the different behaviour of this region, which might appear unexpected for the readership. I leave this decision to the editor and the authors.

- **Thank you for your comments and we agree that the lack of correlation in Southern Europe is because dry and anticyclonic conditions are the average conditions over this part of Europe during Summer. The lack of correlation is also built in by construction through use of the Sousa et al. 2021 algorithm. This identifies sub-tropical ridges and blocking systems that occur north of the subtropical high boundary. Given that most of Southern Europe lies under the sub-tropical high throughout summer, the algorithm does not identify subtropical ridge or blocking conditions there. This can result in little correlation there.**
    - **We have masked grid cells below 40N in Fig. 7 and Fig. 8 for the above reasons and to ensure only physically meaningful results are shown. The justification for this is explained on P6 L191-195 and further highlighted on P23, L436-438.**
- **To further supplement this analysis, we have also carried out the same analysis using blocking criteria only (Eq. 2 and Eq. 3, i.e. the subtropical ridge criteria is ignored). The results are shown in Figure A2 in the appendix. Results are largely similar when considering only blocking criteria though there are differences. Specifically, we see positive correlations between the frequency of blocking days (within events longer than 5 days) with both dry spells and temperatures over larger areas of France and Southern Europe, although the correlations are weaker than in more northern areas. The correlations at more northern latitudes are relatively similar to those from the block-ridge criteria though they can differ**

in magnitude, particularly over Central Europe for dry spells and in Eastern Europe for Temperature. Hence the results can depend on the criteria used for the identification of anticyclonic systems, though the overall message is the same.

- o **To highlight this message, we have decided to include the blocking only analysis within the appendix (Fig. A2) and discussed these results within the text (P24 L486-492 and P30 L579-585).**
- **We acknowledge that the choice of whether to choose the blocking-only algorithm or the combined block-ridge algorithm is subjective. We choose to show results for the latter as correlations are higher with temperature in the assessed regions which provides stronger conclusions. We do however highlight that care should be taken in the choice of algorithm used to detect anticyclonic systems and that the optimum choice depends on the region of interest (P24, L486-492 and P30, L579-585).**

**Minor comments**

a) Abstract: blocking plays quite a central role in the paper but it is hardly mentioned in the abstract. The meaning of AS in not explained (line 27). Please enhance.

- **We note that we already refer to anticyclonic systems (AS) as atmospheric blocking and sub-tropical ridges in the abstract (P1 L25). We also state the results related to AS (P1 L25-33).**

b) line 12: please change "temperatures" to "high temperatures"

- **This has been changed.**

c) line 75: I guess the word "ensemble" is missing, hence "smaller climate model ensemble". Please enhance.

- **Yes, thank you. This has been changed.**

d) line 81: I am not sure this is true. please enhance.

- **We have changed the sentence to be more specific and changed 'climate models' with 'CMIP5'. We would also be grateful to the reviewer to point out papers that we may have missed.**

e) line 234: I wonder why you work with ensemble averages, as this will surely flatten the whole signal and might thus weaken the relationships. Have the authors performed the analysis with all the single ensemble members? Are the results and conclusions similar?

- **Ensemble averages are necessary to ensure each model has equal weighting when calculating a multi-model median, as described in the text (P8, L249). If we combine all individual members, it would give those with larger ensembles more weight.**

- **We have carried out the analysis of dry spells and temperatures for all the single ensemble members (Fig. 2 and Fig. 4) and discussed these results throughout the text. The results show the typical spread within each individual model ensemble is much smaller than the spread of the CMIP5 ensemble. Hence, the results and conclusions are similar for the individual model members.**

f) line 332: please add "largely", hence "vary largely independent"

- **I'm not sure this sentence would read well with this addition, so I prefer to not change it. In any case, it would not change the meaning of the sentence.**

g) line 346ff: I agree this is difficult to explain, and should be related to the reasoning given above in the major comment. The same is true for section 4.4 and 4.5. Please enhance.

- **The reasoning for this was given previously in the text though we have added to this to enhance its description (P16, L355-364). We also note that we have enhanced the description of the reasoning for very long dry spells in southern Europe on P8 L262-264.**

**Reviewer #2**

The presented study compares hot-dry long duration events in CMIP5 data (33 members) to their representation in EOBS data in the period 1976 to 2005. Moreover, the authors investigate the relation to anticyclonic systems and blocking by additionally using ERA5 geopotential height data at the 500 hPa level. The authors improved the analysis based on anticyclonic systems. They now use a blocking/extended ridge identification method instead of their previously used fixed pressure threshold which considerably improves the paper and which was a mayor concern given by one reviewer. Moreover they added more text and literature with respect to the processes related to anticyclonic systems/blocking and their relation to temperature extremes. The dynamics is still not the main focus of the paper, but it does better fit within the scope of the journal now. The paper has some interesting results, but I still have some remarks that should be addressed before publication.

**We thank the reviewer for their valuable time and effort taken to review our manuscript. This is much appreciated. Your constructive comments have certainly improved the manuscript.**

**Main points:**
- From the title I expected to see "large discrepancies". However, by looking at the figures, I do not really see them. Sorry, I also do not get it from reading your text. Do you mean, that the spread of the ensemble is very large (which is the purpose of an ensemble)? Either you need to explain more clearly what you mean by "large discrepancies" or you need to think about another title.

- **Thank you for your comment and we agree that this should be clarified. "Large discrepancies" refer to the large spread from the multiple model simulations within the ensemble. The spread between simulations can in principle arise from internal climate variability and differences across models. It is important to distinguish uncertainty arising from internal climate variability from that arising from model differences. This is highlighted in the manuscript though we have tried to enhance its description (P3 L86-93, P9 L278-284 and P29 L531-533), and added caveats to its interpretation (P29, L537-541).**

- The purpose of this spread is to represent our uncertainties about some aspect and this spread can be large if models cannot accurately represent this aspect. In this analysis, our results (see Fig. 2 and Fig. 4) indicate that a substantial part of the spread across the simulations arises from differences across models. This is because the typical spread across ensemble members of a single model is substantially smaller than the overall spread across model simulations.

- We then show that the misrepresentation of persistent anticyclonic conditions largely explains the differences between climate models in their representation of dry spells and, to a lesser extent, temperature extremes in northern Europe (Fig. 8). Hence, the study contributes to informing the potential for reducing uncertainties in compound events through improving climate model representation of processes causing hot-dry events. Specifically, the representation of anticyclones in climate models could reduce the spread between models, and possibly lead to reduced uncertainty in future climate projections which would be beneficial for planning purposes.

- To clarify these aspects, we have enhanced the text to reflect the information above in the new version of the manuscript (P3 L87-93, P9 L278-284 and P29 L531-541). Furthermore, we have modified the title of the paper from "Large discrepancies in the representation of compound long-duration dry and hot spells over Europe in CMIP5" to "Large spread in the representation of compound long-duration dry and hot spells over Europe in CMIP5". We have also removed the word 'Fundamental' to soften the tone.

- I think that the abstract is a bit misleading. The authors say that there are large discrepancies between CMPI5 and EOBS, but in the paper I see mostly aggreement between the CMIP5 ensemble mean and EOBS. Single models/single model ensembles might be be biased, but the whole ensemble of all 33 CMIP5 models captures relatively well the duration return levels of dry spells (except of Scandinavia and partially the Alpine region). The same is true for maximum temperature: The significant differences found by the authors are mainly on grid points that are partially over the oceans. This has indeed an impact on temperature! It would be interesting, how the results look like if all these points are excluded from the analysis.

- There is agreement between EOBS and the ensemble median but, as described in our previous comments above, there still exists a large spread between models around this median. We argue that this spread is due to differences between models in their representation of anticyclonic systems. If single models are biased, then the ensemble median may achieve the right result for the wrong reasons. Thus, using such models for future projections may produce misleading results if they cannot accurately simulate the important processes.
  - We agree that 'large discrepancies' can be misleading and so we have updated the title as described above. We have also enhanced the text by replacing 'discrepancies' with more appropriate terms such as 'spread' throughout the text.

- With respect to differences between EOBS and CMIP5 temperatures at coastal grid cells. We agree that the coastal effect will have an impact on temperature differences at those locations, as is highlighted already in the manuscript (P13, L308-310). However, this coastal effect does not impact the main conclusions about the spread between models. To demonstrate this, we have calculated the standard deviation between models at all grid cells (Fig. A1). The standard deviation between models is not higher at coastal grid cells than the nearby grid cells in land, meaning the variability or spread between models is not higher

at coastal locations. In fact, the highest standard deviations occur further inland away from the coast. Thus, removing coastal grid cells will not change the main conclusions of this analysis that there is a large spread between models.

  o **To clarify the above we have added the supplementary figure (Fig. A1) showing the inter-model standard deviations and added further discussion of the results around the coastal affects on the analysis (P13 L318-322).**

- The authors improved the method to identify anticyclonic systems by using the mothod of Sousa et al (2021) that identifies extended subtropical ridges as well as blocks, however, the method seems to work only north of about 40 degrees north. Why do you not include the subtropical high south of 40 degrees north, too? For example, you could calculate the geostrophic vorticity from the geopotential field at 500 hPa in order to determine anticyclonic motion south of 40 degrees north. However, at least the data south of 40 degrees north should not be used to draw any conclusions from the anticyclone system analysis.

- **The algorithm produces a low frequency of anticyclonic systems in Southern Europe by design. This identifies sub-tropical ridges and blocking systems that occur north of the subtropical high boundary. This boundary is identified using the criteria in the calculate of $LAT_{MIN}$ (P5, L158-164). Given that most of Southern Europe lies under the sub-tropical high throughout summer, the algorithm does not identify sub-tropical ridge or blocking conditions there.**

  o **Unfortunately, it is not within our scope to implement another algorithm, although this might be interesting as discussed on P5 L132-138. However, as described in response to the first reviewer, we have included results for when only the blocking criteria are applied (Eq. 2 and Eq. 3, i.e. subtropical ridge criteria including $LAT_{MIN}$ are ignored), which is the criteria employed by the often used Tibaldi and Molteni (1990) method. This is shown in Fig. A2 and discussed within the manuscript (P24, L486-492 and P30, L573-580). We have also masked areas below 40N to ensure that only physically meaningful results are shown for results obtained using the Sousa et al. (2021) algorithm. We have also enhanced the discussion around the interpretation of these results for Southern Europe (P24, L486-492 and P31, L579-585).**

- Fig. 7: Please add a difference plot: It would be easier to see differences between CMIP5 and ERA5. My naive interpretation of the plot is that CMIP5 shows more or less the same anticyclone system frequencies or at least the same tendency as ERA5. That is why I do not agree with your sentence (e.g. in the abstract: "Overall, there are large discrepancies in the representation of long-duration dry and hot events that are due to fundamental errors in the representation of large-scale anti-cyclonic systems in certain parts of Europe" --> I do not see the "fundamental errors", please elaborate.

- **We agree that the differences are hard to see and so we have added a difference plot to Fig. 7.**

- The correlation coefficients in Fig. 8 could be biased by the outlier that can be seen in Figs. 8(c), (d), and (e) (in the top right corner of the plots). Please test if the correlation holds if you exclude those

outlier points. Maybe you can try this systematically by excluding point by point the ones with the largest distance to the point cloud center.

- **We agree that the Pearson correlation coefficient may be overly influenced by an outlier. We therefore have changed this figure and used the Spearman rank correlation coefficient which will be much less sensitive to outliers. For consistency, we have also changed Figure 6 to show the Spearman correlation instead of the Pearson correlation. We have not looked at removing outlier points as the Spearman correlation will not be very sensitive to this and we feel that the scatter plots effectively show that there is a clear relationship between the frequency of anticyclonic systems and dry spells as well as temperatures.**

- Can you hypothesize why the correlation of dry spells and anticyclones is largely positive directly north of the maximum anticyclone frequency? (Fig. 8)

- **Thank you for this comment. We are not sure about this and there could be a number of reasons. Firstly, the highest frequency occurs over the Alps. It is possible that the poor representation of orography in coarse resolution models will break the association between anticyclones and dry spells, or indeed that anticyclones have a reduced effect on rainfall over high elevation areas. It may also be a simple consequence of a naturally higher frequency of anticyclones closer to 40N as ridges extruding from the subtropical belt will influence these areas more frequently, i.e. there may not be a physical relationship to explain this. For instance, there are no high correlations over northern Finland which lies above a peak in anticyclone frequency.**

**Minor points:**
- The abstract needs to be checked and maybe updated. Moreover there are abbreviations (e.g. AS) that are not properly introduced. And please add your study period.

- **Thank you for pointing this out. We have removed 'AS'. We have also updated the abstract to better reflect the results. For example, 'large discrepancies' is replaced with more appropriate terms such as 'spread between models'.**

- why do you use in section 4.5 the multi-model median and not the multi-model mean?

- **Thank you for noting this. The median is used here as it better represents the centre of the distribution or ensemble. This is preferred to the mean as the mean can be overly influenced by single outlier models. We also note that we have used the median throughout the analysis for dry spells and temperatures for the same reason. These were mistakenly referred to as the mean, this has now been updated throughout the text and in the methods (P8 L248-252).**

- Figs. 1,3, 5,6,8 Please try to change the divergent colormaps, because they are partially misleading: I plotted you paper on a b/w printer and could not identify if the values are positive or negative. Maybe you can add a solid and dashed line for positive /negative or use another colormap if possible.

- **Thank you for this suggestion. We have changed the divergent colormap in Figure 1a,b for duration RLs to a continuous colormap to avoid confusion with plots showing differences and ratios. However, it is necessary to have divergent colormaps for showing differences and ratios. And it is difficult to account for those needing to print b/w, we feel being able to view the figure online should suffice. We have looked at adding contours but the grid is so coarse that the contours take away from the figures so we prefer not to add these.**

---

## Author Response (AR3)

**Editor comments:**

Dear Colin Manning and co-authors,

Thank you for fully addressing the comments of all reviewers and myself. Considering this relevant topic, your revised work with the added focus on anticyclones and blocking now provides an important contribution to understanding model spread.

I am very happy to accept your manuscript for publication in WCD. Congrats!

Please consider these two technical corrections and provide a corrected version.
Fig. 7b: correct the caption to reflect the change you made to this panel
Fig. A2: remove the remark about the grey area from the caption

Kind regards,
Shira

**Response to Editor:**

**Dear Shira,**

**Many thanks for your kind words and positive feedback on our work. And thank you also for your valuable time taken in overseeing the review of this manuscript, it is much appreciated. As requested, we have made the amendments to the figure captions, and we thank you for noticing this detail.**

**Kind regards,**

**Colin**